ⓐ | **Open Peer Review** | Genetics and Molecular Biology | Research Article

# Unraveling the transcriptional features and gene expression networks of pathogenic and saprotrophic *Ophiostoma* species during the infection of *Ulmus americana*

**Thais C. de Oliveira,**[1,2] **Nastasia J. Freyria,**[3] **Jorge Luis Sarmiento-Villamil,**[1,2,4] **Ilga Porth,**[1,2] **Philippe Tanguay,**[5] **Louis Bernier**[1,2]

**ABSTRACT**    American elm (*Ulmus americana*), highly prized for its ornamental value, has suffered two successive outbreaks of Dutch elm disease (DED) caused by ascomycete fungi belonging to the genus *Ophiostoma*. To identify the genes linked to the pathogenicity of different species and lineages of *Ophiostoma*, we inoculated 2-year-old *U. americana* saplings with six strains representing three species of DED fungi, and one strain of the saprotroph *Ophiostoma quercus*. Differential expression analyses were performed following RNA sequencing of fungal transcripts recovered at 3- and 10-days post-infection. Based on a total of 8,640 *Ophiostoma* genes, we observed a difference in fungal gene expression depending on the strain inoculated and the time of incubation in host tissue. Some genes overexpressed in the more virulent strains of *Ophiostoma* encode hydrolases that possibly act synergistically. A mutant of *Ophiostoma novo-ulmi* in which the gene encoding the *ogf1* transcription factor had been deleted did not produce transcripts for the gene encoding the hydrophobin cerato-ulmin and was less virulent. Weighted gene correlation network analyses identified several candidate pathogenicity genes distributed among 13 modules of interconnected genes.

**IMPORTANCE** *Ophiostoma* is a genus of cosmopolitan fungi that belongs to the family Ophiostomataceae and includes the pathogens responsible for two devastating pandemics of Dutch elm disease (DED). As the mechanisms of action of DED agents remain unclear, we carried out the first comparative transcriptomic study including representative strains of the three *Ophiostoma* species causing DED, along with the phylogenetically close saprotrophic species *Ophiostoma quercus*. Statistical analyses of the fungal transcriptomes recovered at 3 and 10 days following infection of *Ulmus americana* saplings highlighted several candidate genes associated with virulence and host-pathogen interactions wherein each strain showed a distinct transcriptome. The results of this research underscore the importance of investigating the transcriptional behavior of different fungal taxa to understand their pathogenicity and virulence in relation to the timeline of infection.

**KEYWORDS**    transcriptome, differential gene expression, *Ophiostoma*, pathogenicity, host-pathogen interaction, Dutch elm disease

Address correspondence to Thais C. de Oliveira, thais.campos-de-oliveira.1@ulaval.ca, or Louis Bernier, louis.bernier@sbf.ulaval.ca.

The authors declare no conflict of interest.

See the funding table on p. 20.

Human activities have resulted in a dramatic increase in the global movement of fungal tree pathogens, which have spread into both natural forests and tree plantations on several continents, hereby threatening biodiversity and causing important economic losses (1–4). *Ophiostoma ulmi* and *Ophiostoma novo-ulmi* (Ascomycota, Ophiostomatales) are the causal agents of the highly destructive Dutch elm disease (DED), which impacted field and urban elm populations worldwide (5). Infection of a susceptible individual rapidly leads to vascular wilt, dieback, and death of the host (5).

Elm bark beetles belonging to genera *Scolytus* and *Hylurgopinus* vector the pathogen and allow it to penetrate the host vascular system when they feed on healthy elms (6, 7). Six taxa, including three distinct species, are recognized in the DED fungi. The moderately aggressive *O. ulmi* (OU) was responsible for the first pandemic, which began in the early 1900s (8) and persisted until the 1950s to the 1980s depending on the location. The second, ongoing pandemic is caused by the highly aggressive *O. novo-ulmi* (1, 9), which progressively replaced *O. ulmi* in most areas and includes two subspecies designated *novo-ulmi* (ONU) and *americana* (AME) (9, 10). Two distinct genetic lineages, AME1 and AME2, have been reported in subsp. *americana* (11–13). A third species, *Ophiostoma himal-ulmi* (OHU), was recovered from symptomless *Ulmus wallichiana* in the Indian Himalayas and shown to be pathogenic toward European elm varieties; it is thus considered a DED fungus (14). All DED fungi exhibit yeast-mycelium dimorphism and the *in vitro* response of this trait to external stimuli varies according to individual strain (15, 16). Furthermore, physiological, molecular, and genomic data have shown the occurrence of *O. ulmi*-type DNA introgression into the *O. novo-ulmi* genome (13, 17–19), as well as the emergence of hybrid swarms resulting from sexual crosses between subsp. *novo-ulmi* and *americana* individuals (20).

The DED fungi belong to a group of closely related species (21), which also includes saprobes causing sapstain. One of them, *Ophiostoma quercus* (OQ), has a cosmopolitan distribution (22–24) and was suggested to be the species from which the DED fungi evolved following interspecific hybridization and/or secondary speciation (25). Some *O. quercus* mutants transformed with the *O. novo-ulmi* gene encoding the hydrophobin cerato-ulmin (CU) were reported to be pathogenic to Commelin elm (26).

The meteoric growth of genomic sequencing technologies in the last decades (27) offers unprecedented opportunities for investigating further the biology of plant pathogens and their interactions with their hosts. For instance, high-quality reference genomes have been obtained for fungal wilt pathogens, including *Fusarium oxysporum* f. sp. *lycopersici* (28), *Verticillium albo-atrum* and *V. dahliae* (29), and the DED fungi *O. ulmi* (30) and *O. novo-ulmi* (31). The availability of a fully assembled and well-annotated genome for *O. novo-ulmi* subsp. *novo-ulmi* (32) has facilitated genome-wide analyses of population dynamics of DED fungi (13) and transcriptomic profiling of the yeast-mycelium growth dynamics of *O. novo-ulmi in vitro* (33, 34). Comparative *in silico* analyses also showed that a fujikurin-like OpPKS8 biosynthetic gene cluster found in *O. ulmi* and *O. novo-ulmi* was likely acquired through horizontal gene transfer from a taxon belonging to a different family in the Ascomycota and might contribute to the pathogenicity of the DED fungi (35).

Studies of DED fungi based on the -omics *in planta* lag behind those carried out *in vitro* or *in silico*. Pioneering works on elm-*O. novo-ulmi* interactions (36, 37) allowed the recovery of transcripts for less than 200 fungal genes and were therefore not very informative regarding the biology of the pathogen. Nigg et al. (38) presented the first comprehensive analysis of transcriptomes of *O. novo-ulmi* subsp. *americana* following inoculation of one susceptible and one resistant (Valley Forge) clone of *Ulmus americana*. In the work described herein, we compared the transcriptomes of seven *Ophiostoma* strains, which had been inoculated to *U. americana* saplings. The strains represented all taxa of DED fungi, along with the saprobe *O. quercus* and one mutant of *O. novo-ulmi* subsp. *novo-ulmi* in which gene *ogf1* had been deleted (ΔOgf1). Gene *ogf1* encodes a putative homolog of the *Magnaporthe oryzae* GPF1 fungal-specific Zn2Cys6 transcription factor, which is required for normal growth, conidia germination, appressorium formation, and virulence (39–41). We observed marked differences among the *Ophiostoma* transcriptomes and identified several candidate genes that might be involved in the parasitic fitness of the DED pathogens.

## RESULTS

### External symptoms in American elm saplings inoculated with *Ophiostoma* spp.

At 3 days post-inoculation (dpi), low levels of leaf wilting were observed in most saplings inoculated with ONU or ΔOgf1, whereas at 10 dpi, moderate levels of wilting were observed in saplings inoculated with OU, ONU, AME2, or ΔOgf1 (Table S1.1). In additional tests of the effect of inoculum concentration on the virulence of wild-type (WT) strain ONU and its *Δogf1* mutant, both strains induced similar levels of leaf symptoms (including wilting, yellowing, browning, or abscission) when inoculated at higher doses (50,000 or 100,000 cells), whereas strain ΔOgf1 showed a partial (25%) but significant ($P <$ 0.05) reduction in virulence when saplings were inoculated at lower doses ($3.12 \times 10^3$ or $1.25 \times 10^4$ cells; Tables S1.2 to S1.4). Mutant ΔOgf1 was also significantly less virulent ($P < 0.05$) than strain ONU on Golden Delicious apples, based on the diameter of necroses developing around the inoculation point (Table S1.5).

### Overview of the *Ophiostoma* spp. transcriptomes *in planta*

In total, 2 TB of data were collected after sequencing, processing, and mapping of 68 RNA samples recovered from *U. americana* saplings inoculated with a strain of *Ophiostoma* or injected with sterile water (Table S2). Since not all genomes and functional annotations were publicly available for the fungal taxa assayed in this work, we used the well-characterized *O. novo-ulmi* subsp. *novo-ulmi* strain H327 genome sequence consisting of 8,640 genes (32) as our model for mapping and analyzing all fungal transcripts. The *Ophiostoma* transcriptomes included 9,268,611 raw reads at 3 dpi and 7,073,182 raw reads at 10 dpi (Table S2). A total of ca. 280,000 uniquely mapped reads were obtained for each fungal strain (Table S2). The numbers of reads for all ONU H327 orthologs in *Ophiostoma* spp. are shown in Table S3. Fungal transcripts recovered from water-injected elms were considered to represent conserved genes of fungal endophytes of *U. americana* (see Discussion) and therefore removed from further analyses [principal component analysis (PCA), Heatmap, Gene Ontology (GO), Kyoto Encyclopedia of Genes and Genomes (KEGG), Carbohydrate-Active enZYmes (CAZy), and Pathogen-Host Interactions database (PHI-base)].

Depending on the strain, the top 10 most-expressed genes were represented by an expression level of 3 to 3,263, with lower numbers recorded for OQ (Table S4). Gene *OnuG5955* encoding an alcohol oxidase was the most highly expressed gene in all strains except OQ. This gene was one of 90 *O. novo-ulmi* genes expressed only *in planta* in previous transcriptomics studies (38). Transcripts for 82 of the remaining 89 genes were detected with counts per million > 4 (cpm > 4) in our data set (Table S5). Genes from DED-causing *Ophiostoma* species usually exhibited comparable expression values, with a few notable exceptions, such as *OnuG7311* (protein TOXD), which was more expressed in ONU, *OnuG5670* (exopolygalacturonase) not detected in OHU, and *OnuG4739* (flavin-containing monooxygenase) more highly expressed in OU. Gene *OnuG1537* was not expressed (cpm < 4) in mutant ΔOgf1 (Table S3).

### Differential expression analysis

The differential expression analysis of transcriptomes at 3 and 10 dpi (Fig. 1A) showed different patterns among *Ophiostoma* strains. Strains OU, AME2, and OHU had intermediate to high total numbers of differentially expressed genes (DEGs) ($n$ = 324–603) and these were more abundant at 3 dpi than at 10 dpi. Strains ΔOgf1 and AME1 had moderately high numbers of DEGs ($n$ = 333 and 353, respectively) and these were observed in higher numbers at 10 dpi than at 3 dpi. A similar type of distribution was observed in OQ but, in this case, the total number of DEGs was low ($n$ = 114). In the case of ONU, the 204 DEGs were distributed roughly equally between the 3- and 10-dpi subsets. The 12 DEGs at 3 dpi recorded in water-injected elms were considered to represent conserved genes of fungal endophytes.

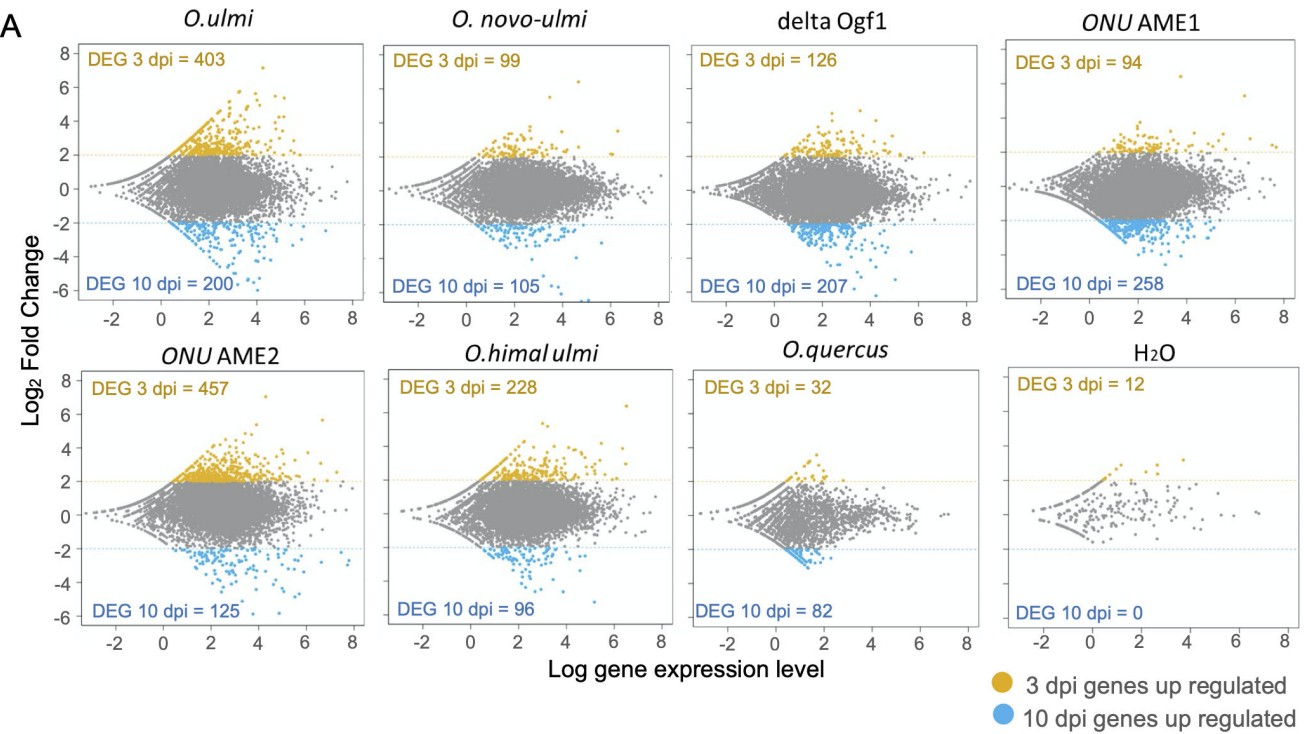

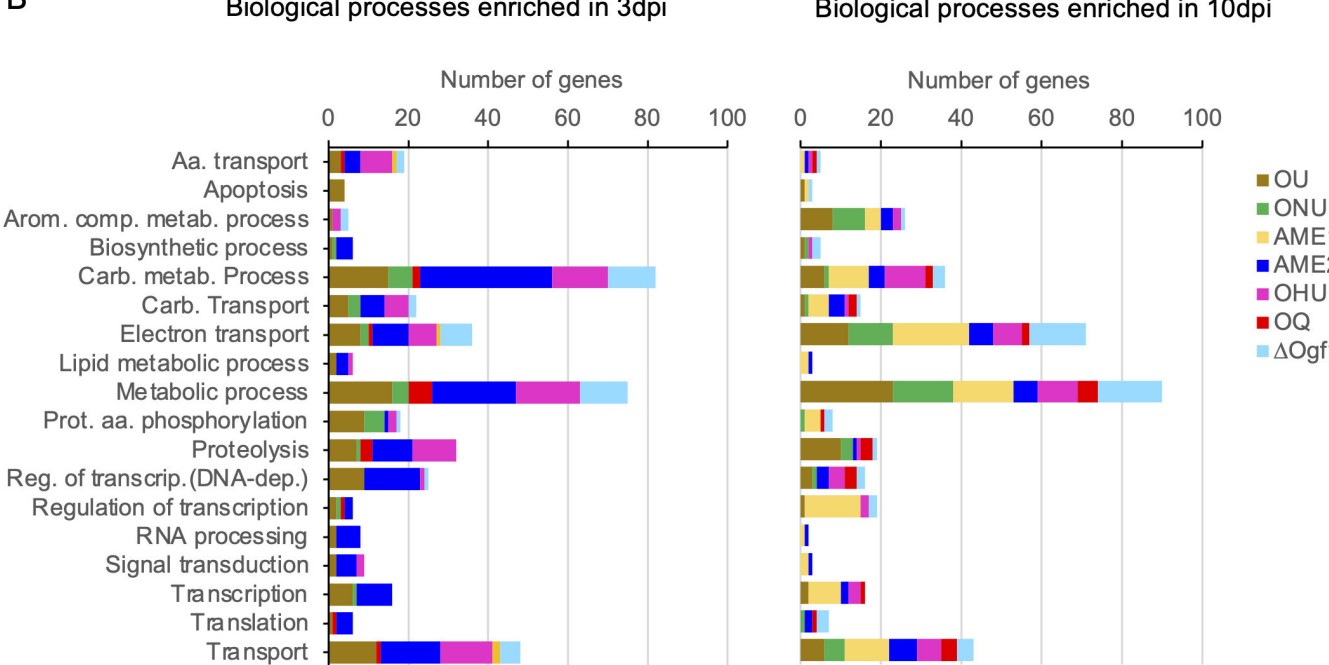

**FIG 1** Differentially expressed *Ophiostoma* spp. (OU, ONU, AME1, AME2, OHU, OQ, and ΔOgf1) genes during interaction with *Ulmus americana*. (A) MA plot of all *Ophiostoma genes* for which transcripts were detected including genes that were overexpressed at 3 and 10 dpi. Read counts were normalized using DESeq2 package in R with $log_2FC > 2$ and scatter plot visualization. Fungal transcripts recorded in water-injected elms represent conserved genes of fungal endophytes. (B) Gene ontology terms for the top 18 biological processes that were significantly enriched in *Ophiostoma* (OU, ONU, AME1, AME2, OHU, OQ, and ΔOgf1) colonizing *U. americana* at 3 dpi (left) and 10 dpi (right).

Principal component analysis of the transcription data set for the seven *Ophiostoma* strains clearly distinguished OQ from all specimens of DED fungi (Fig. 2). In general, data obtained at 3 dpi tended to be clustered toward the positive side of PC2, as was the case

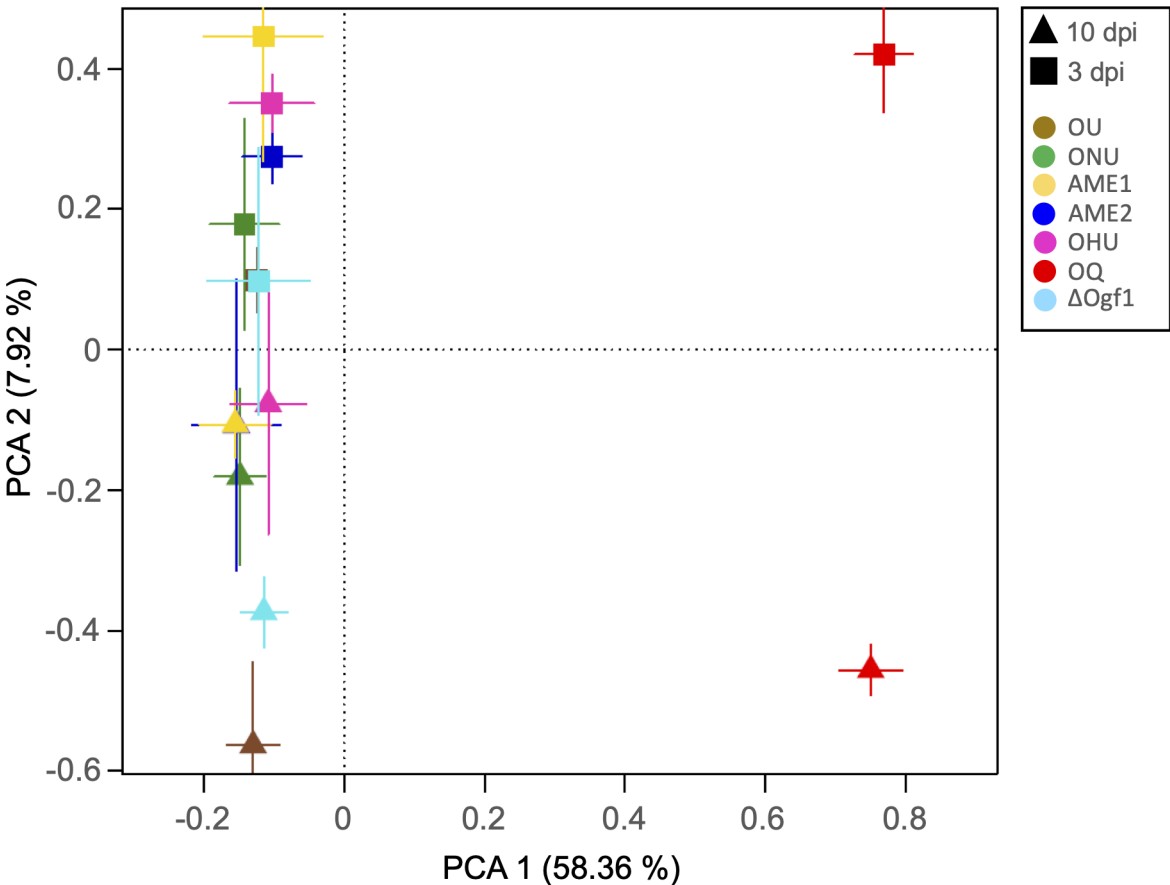

**FIG 2** Principal component analysis of the complete transcription data set for the seven *Ophiostoma* spp. strains (OU, ONU, AME1, AME2, OHU, OQ, and ΔOgf1) inoculated to *Ulmus americana* saplings. The symbol for AME2 at 10 dpi is hidden behind the symbol for AME1 at 10 dpi, and the symbol for OU at 3 dpi is hidden behind the symbol for ΔOgf1 at 3 dpi. Each group is represented as the average position of all samples within the group and the standard errors of each PC as the error bars.

for all replicates for strains OU, AME2, and OHU. Although strains ONU and ΔOgf1 differed only by a mutation at a single locus, their transcription patterns tended to be distinct from each other. When DEGs were compared among *O. novo-ulmi* strains, the Venn diagram analysis yielded contrasting results (Fig. S1). At 3 dpi, strains ΔOgf1 and AME2 shared the highest number of DEGs ($n = 30$), followed by the AME1-AME2 pair ($n = 29$), whereas only 12 DEGs were common to strain ΔOgf1 and its wild-type ONU parental. At 10 dpi, the number of DEGs shared by ΔOgf1 and AME2 dropped to only six, whereas ΔOgf1 and AME1 now shared the highest number of DEGs ($n = 38$).

We conducted comparative heatmap analyses of the top 50 DEGs in OU and ONU at 3 and 10 dpi, following variance stabilizing transformation (Fig. S2). At 3 dpi, we identified four DEGs (*OnuG0653*, *OnuG3998*, *OnuG8505*, and *OnuG8458*) encoding small unknown proteins of 322, 222, 283, and 296 amino acids, respectively, whereas a single DEG (*OnuG5230*) for a small unknown protein of 76 aa was observed at 10 dpi. These five genes were upregulated in OU. Furthermore, four of the eight genes in the OpKS8 putative fujikurin-like cluster (*OnuG 7305*, *7306*, *7311,* and *7312*; see Table S6 for annotation) were upregulated in ONU at 10 dpi (Fig. S2). These genes, along with *OnuG7310*, were downregulated in mutant ΔOgf1 at 10 dpi compared to ONU (Fig. S2). In addition, the ΔOgf1 mutant did not express the *OnuG4296* gene encoding cerato-ulmin.

## GO enrichment analysis

Comparative analysis of GO terms for "biological processes" that were significantly enriched showed that "carbohydrate metabolic process," "metabolic process," and "transport" were the terms that included the highest numbers of DEGs at 3 dpi (Fig. 1B). Strains AME2, OU, and OHU accounted for most DEGs in "carbohydrate metabolic process" ($n$ = 33, 15, and 14, respectively), "metabolic process" (21, 16, and 16, respectively), and "transport" (15, 12, and 13, respectively) at 3 dpi. In contrast, at 10 dpi, the numbers of AME2 and OU DEGs linked to "carbohydrate metabolism" dropped drastically to four and six, respectively. Moreover, at 10 dpi, the three terms with the most DEGs were "metabolic process," "electron transport," and "transport." Most DEGs in "metabolic process" originated from OU ($n$ = 23), ΔOgf1 ($n$ = 16), and ONU and AME1 ($n$ = 15 each). Top DEG contributors to "electron transport" were AME1, ΔOgf1, and OU (19, 14, and 12, respectively), whereas most DEGs in "transport" were found in AME1 ($n$ = 11), AME2 ($n$ = 7), and OU and OHU ($n$ = 6 each). In the case of saprobe OQ, the highest numbers of DEGs were recorded at 10 dpi in "metabolic process" and "transport," with five DEGs each.

Analysis of functional annotations for the top 20 GO terms in "molecular function" and "biological process" and all 15 terms detected in "cellular components" confirmed trends observed for global differential expression: the highest numbers of DEGs at 3 dpi were recorded in strains OU, AME2, and OHU, whereas more DEGs were observed in strains ΔOfg1, AME1, OQ, and, to a lesser extent, ONU at 10 dpi (Fig. 3). Furthermore, the global gene expression pattern of the AME2 strain was more similar to that of the OU than to the phylogenetically closer AME1 strain. In "molecular function," we observed high numbers of DEGs linked to "oxidoreductase activity" (for instance, 26 DEGs in OU at 10 dpi and 26 DEGs in AME2 at 3 dpi) and to "hydrolase activity" (notably 35 DEGs in AME2 at 3 dpi). At 3 dpi, DEGs linked to "zinc ion binding," "ATP binding," "DNA binding," and "nucleic acid binding" terms were markedly more abundant in OU and AME2. No DEGs linked to "cellulase activity" were detected in OQ, OU, and ONU, although they were observed in mutant ΔOgf1 at 10 dpi. The latter strain had no DEGs linked to the term "transcription factor activity." In "biological processes" term, mutant ΔOgf1 was once again characterized by the absence of DEGs for the term "transcription." In "cellular components," we observed more genes upregulated linked to the term "membrane" at 3 dpi (OU, AME2, and OHU), "integral to membrane" (OU 3 dpi, AME1 10 dpi, AME2 3 and 10 dpi, and OHU 10 dpi), "nucleus" at 3 dpi (OU and AME2), and "intracellular" at 3 dpi (OU and AME2).

## KEGG analyses

When KEGG pathway annotations were subjected to PCA (Fig. 4A), the component 1 axis separated OQ from all other *Ophiostoma* samples, whereas component 2 separated transcriptomes at 3 dpi from those at 10 dpi. According to PC1, the 3-dpi data set was more homogenous than the 10-dpi data set in which transcriptomes of ΔOgf1 and OHU had diverged from the others. More detailed analyses of *Ophiostoma* DEGs (Fig. 4B) showed more upregulated genes linked to "biosynthesis of secondary metabolites" at 3 dpi, except in AME1 and OQ where no upregulated gene was observed. Strains OU, AME2, and OHU presented more upregulated genes at 3 dpi compared to other strains.

## Analysis of CAZyme-encoding genes

A PCA of all *Ophiostoma* gene reads with orthologs in the CAZy database confirmed that gene expression in OQ differed strongly from that in other *Ophiostoma* species and lineages (Fig. 5A). Expression patterns of lineage AME1 were also well separated from those of other *Ophiostoma* strains investigated. Within each strain, DEG data at 3 and 10 dpi generally clustered, and this was particularly evident in ONU and AME2. The notable exception was OHU in which data at 3 dpi overlapped with 10 dpi data for OU. We documented for the first time the possible action of hydrolases linked to β-glucose

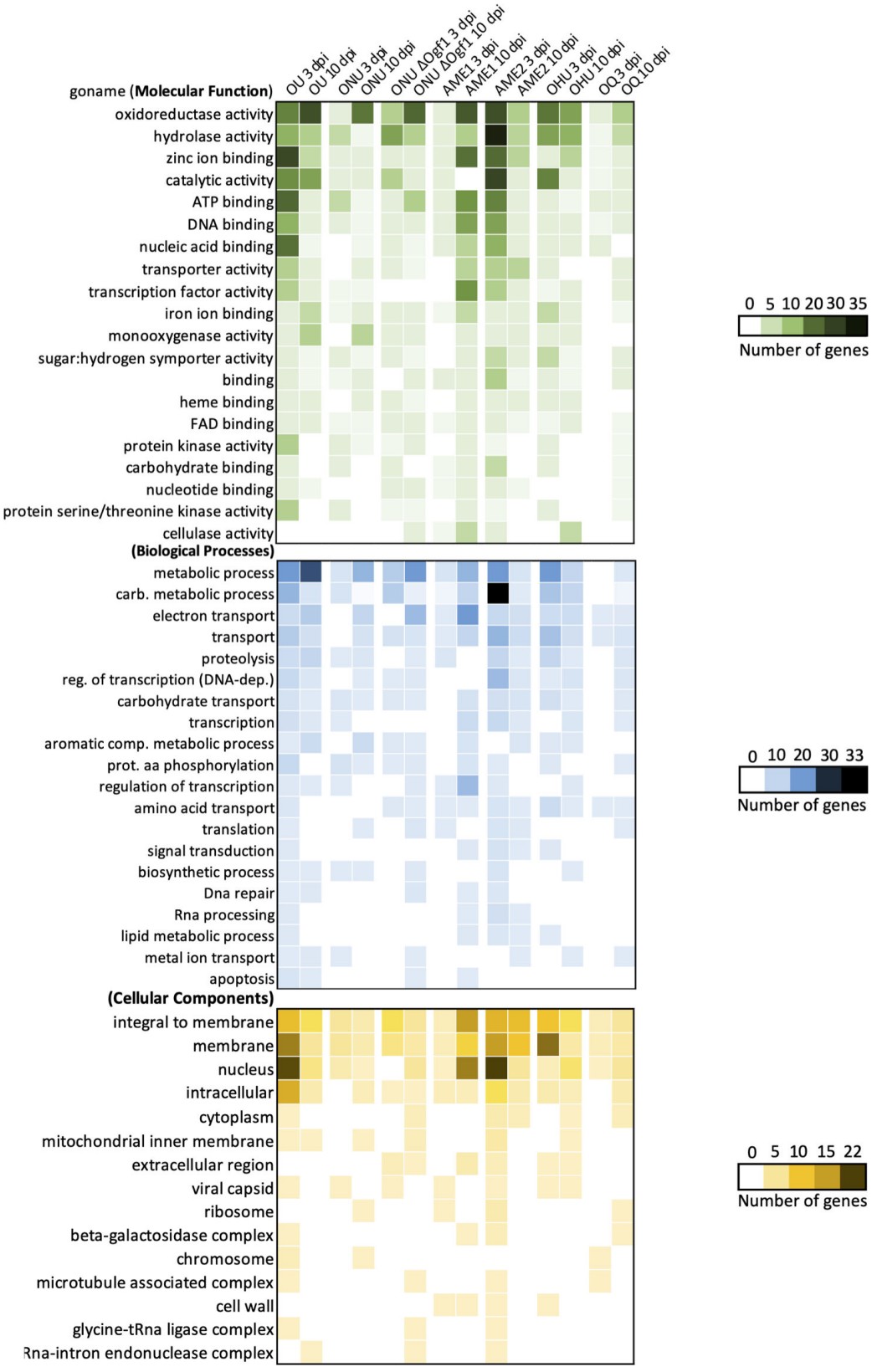

**FIG 3** Functional annotation analysis based on the Gene Ontology of genes expressed at 3 and 10 dpi in *Ophiostoma* spp. strains (OU, ONU, ΔOgf1, AME1, AME2, OHU, and OQ) inoculated to *Ulmus americana*. Heatmap analysis based on the number of genes in the top 20 GO terms in "molecular function" (in green) and "biological processes" (in blue) and all genes contained in "cellular components" (in yellow).

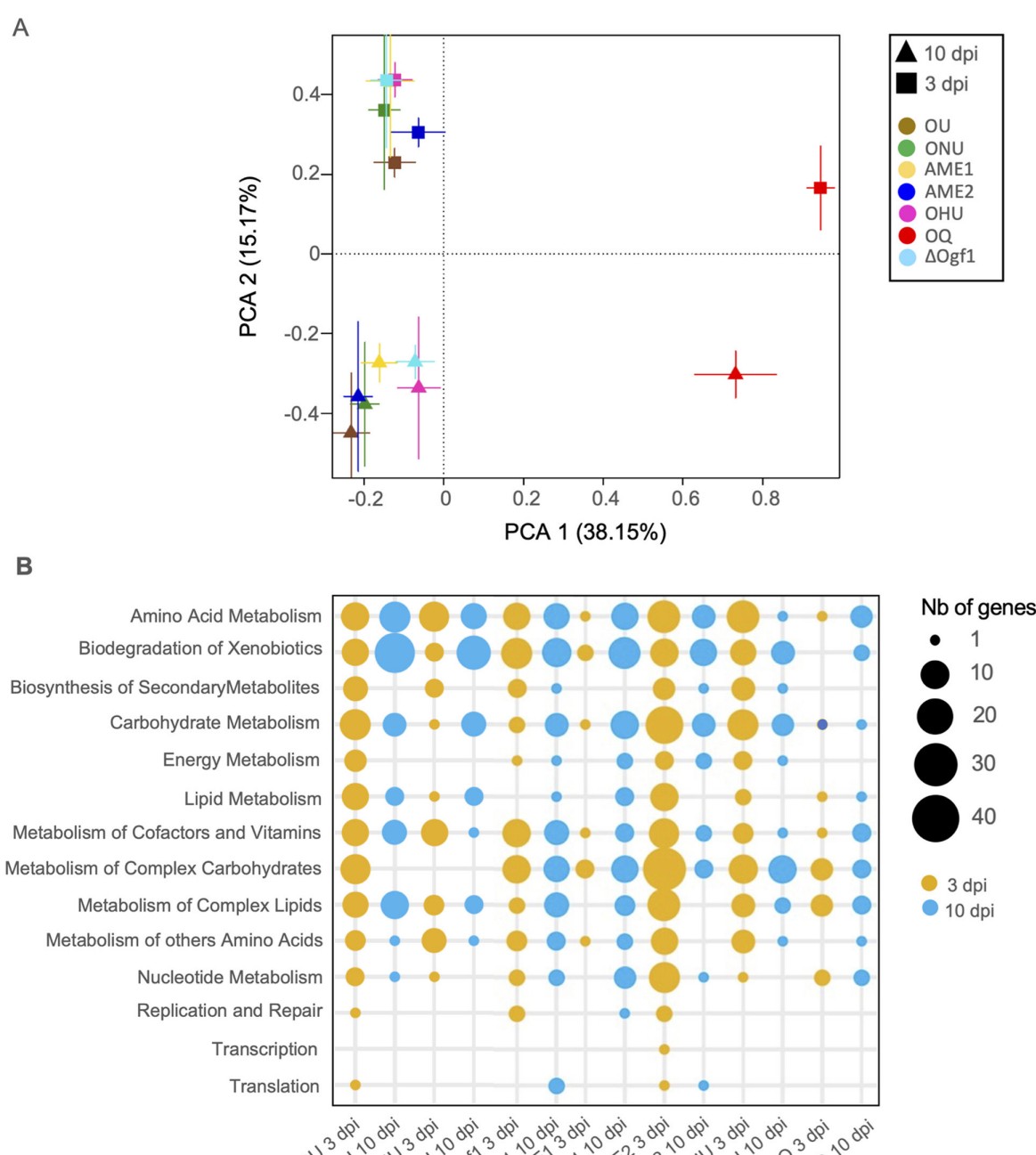

**FIG 4** Functional annotation analysis based on the Kyoto Encyclopedia of Genes and Genomes of genes expressed at 3 and 10 dpi in *Ophiostoma* spp. strains (OU, ONU, ΔOgf1, AME1, AME2, OHU, and OQ) inoculated to *Ulmus americana*. (A) Principal component analysis of the predicted functional features in KEGG. Each group is represented as the average position of all samples within the group and the standard errors of each PC as the error bars. (B) Number of representative KEGG DEGs among *Ophiostoma* taxa and strains.

digestion (Fig. 5B) in some of the DED fungi. We thus observed that gene *OnuG0547* (GH6) was DEG at 3 dpi in OU and AME2, and DEG at 10 dpi in AME1 and OHU; genes *OnuG0647* and *OnuG5654* (GH7) were both DEG at 3 dpi in AME2, whereas the former was DEG at 10 dpi in OHU; and finally, gene *OnuG0647* (GH45) was DEG at 3 dpi in ΔOgf1 and DEG at 10 dpi in AME1 and AME2. Overall, AME2 had the largest number of DEGs for CAZymes (*n* = 56), with most of them (*n* = 48) observed at 3 dpi. Furthermore, 39 of the 56 DEGs encoded a protein expected to have a signal peptide (32). On the other hand, AME1 had 21 DEGs for CAZymes, including 18 DEGs at 10 dpi. The saprobe OQ had the

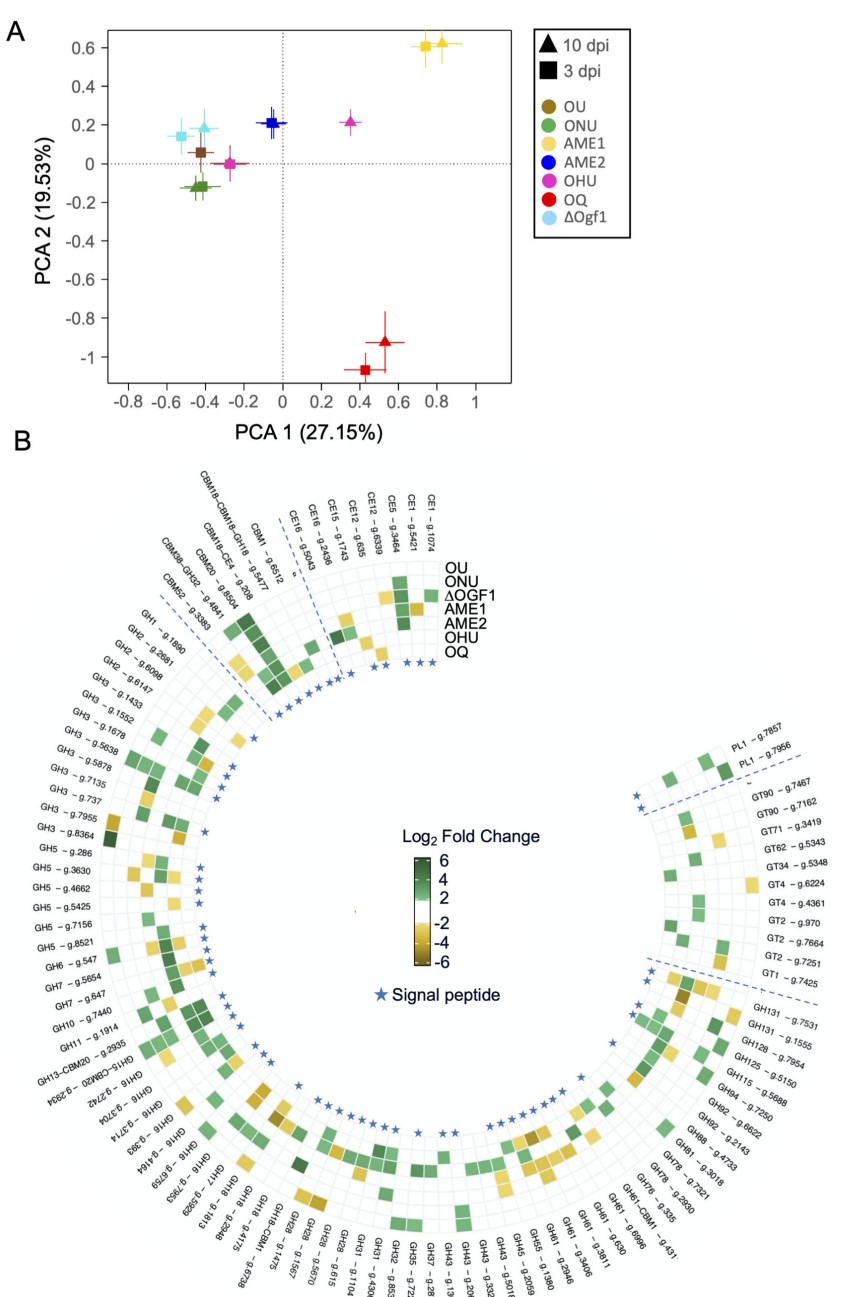

**FIG 5** Functional annotation analysis based on the Carbohydrate-Active enZYmes database of genes expressed at 3 and 10 dpi in *Ophiostoma* spp. strains (OU, ONU, ΔOgf1, AME1, AME2, OHU, and OQ) inoculated to *Ulmus americana*. (A) PCA analysis of *Ophiostoma* CAZyme-encoding genes. Each group is represented as the average position of all samples within the group and the standard errors of each PC as the error bars. (B) Circular heatmap for differentially expressed genes (Log2FC > 2) divided into different families: carbohydrate esterases (CE), glycoside hydrolases (GH), carbohydrate-binding modules (CBM), glycosyl transferases (GT), and polysaccharide lyases (PL). The initials are followed by the gene number represented. DEGs at 3 dpi have a positive FC value and are shown in green; DEGs at 10 dpi have a negative FC value and are shown in yellow. Genes marked with a blue star code for a CAZyme with a signal peptide.

lowest number of DEGs (*n* = 6), split evenly between 3 and 10 dpi. Three of the proteins encoded by OQ DEGs were expected to include a signal peptide.

## Analysis of Pathogen-Host Interactions database

We detected 498 DEGs with orthologs in the PHI-base: 287 at 3 dpi and 211 at 10 dpi. PCA analyses (Fig. 6A) showed a clear separation between OQ and the other species along PC1, and a separation between the 3- and 10-dpi data sets along PC2. Overall, the organisms clustered more closely at 10 dpi than at 3 dpi, apart from OQ. The terms "unaffected pathogenicity" and "reduced virulence" were the most represented in the *Ophiostoma* data set, notably in strains OU, AME2, and OHU at 3 dpi (Fig. 6B). Only one DEG was observed for each of the terms "effector" and "enhanced antagonism" in OU at 3 and 10 dpi, respectively.

## Weighted gene correlation network analyses

We used the weighted gene correlation network analyses (WGCNA) method to identify 13 color-coded modules representing clusters of highly interconnected fungal genes with similar expression changes during infection of *U. americana* saplings (Fig. 7). These modules were obtained through soft thresholding of pairwise correlations between gene expression data, which emphasizes high correlations at the expense of low correlations (42). Within each module, we identified hub genes, i.e., genes that had the most interactions with other genes and are therefore expected to play an important role in gene regulation and biological processes (43). The number of genes in each module was highly variable, with the Turquoise module including 6,794 genes, whereas other modules contained from 53 to 438 genes (Fig. 7). Statistically significant module-trait relationships were observed for all modules, except for the Tan module. This included the Gray module in which gene expression was strongly positively correlated ($P < 0.001$) with the water-injection treatment. All other modules, apart from modules Purple and Tan, were negatively correlated with this treatment. Two modules, Greenyellow and Green ($P < 0.01$ and $P < 0.001$, respectively), were positively correlated with OU. The Purple and Turquoise modules were correlated with ONU ($P < 0.01$ and $P < 0.05$, respectively), and Turquoise module included 69 of the 99 DEGs at 3 dpi and 80 of the 105 DEGs at 10 dpi in this taxon. The Blue module was strongly positively correlated with mutant ONU ΔOgf1 ($P < 0.001$). Module Magenta was positively correlated with AME2 ($P < 0.05$), whereas no module was correlated with the AME1 lineage. As for the modules Black and Pink, they both were positively correlated with OHU ($P < 0.001$). In contrast, only negative correlations were significant ($P < 0.05$ to $0.001$) between modules ($n = 8$) and OQ. When global expression data were compared between the 3- and 10-dpi data sets, the Brown module included genes that were represented by significantly ($P < 0.05$) higher numbers of transcripts at 3 dpi.

## Selected module-specific co-expression network analysis

Co-expression networks were constructed with data from the Purple, Green, and Brown WGCNA modules, respectively. The Purple network, which was correlated with ONU, included the entire set of genes in the fujikurin-like cluster (OpPKS8), which exhibited first-neighbor linkage among them and were all upregulated at 10 dpi. In addition, gene *OnuG7303* encoding a sterigmatocystin 8-O-methyltransferase was strongly upregulated at 10 dpi ($\log_2 FC = -6.45$) and showed a first neighborhood connection with this cluster. The Purple network also included four genes encoding secreted proteins: *OnuG4704* (unknown protein), *OnuG1520* and *OnuG8550* (probable cytoplasmic effector and apoplastic effector, respectively), and *OnuG2969* (choline dehydrogenase). The two genes coding for probable effectors were not DEGs in ONU (Fig. 8).

In the Green module network, which was correlated with OU, the hub gene was *OnuG6492*, which encodes an ortholog of the zinc-responsive transcriptional regulator ZAP1 and displayed first neighborhood linkage with a subgroup of 20 genes, including 16 genes that were upregulated at 3 dpi. No gene encoding a secreted protein was found within the subgroup but six such genes were identified elsewhere in the network. The latter included *OnuG6759* coding for a secreted glucosidase. The other five genes

A

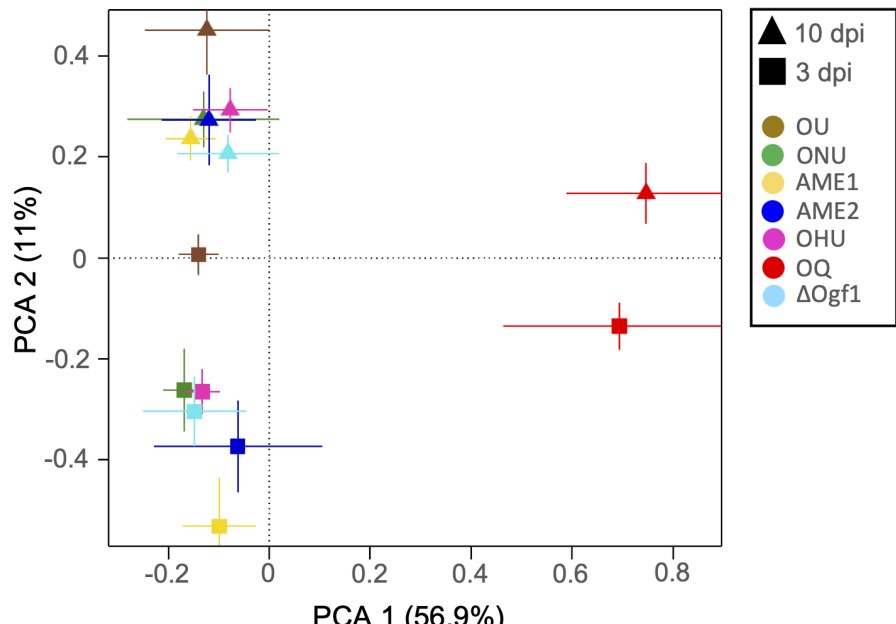

B

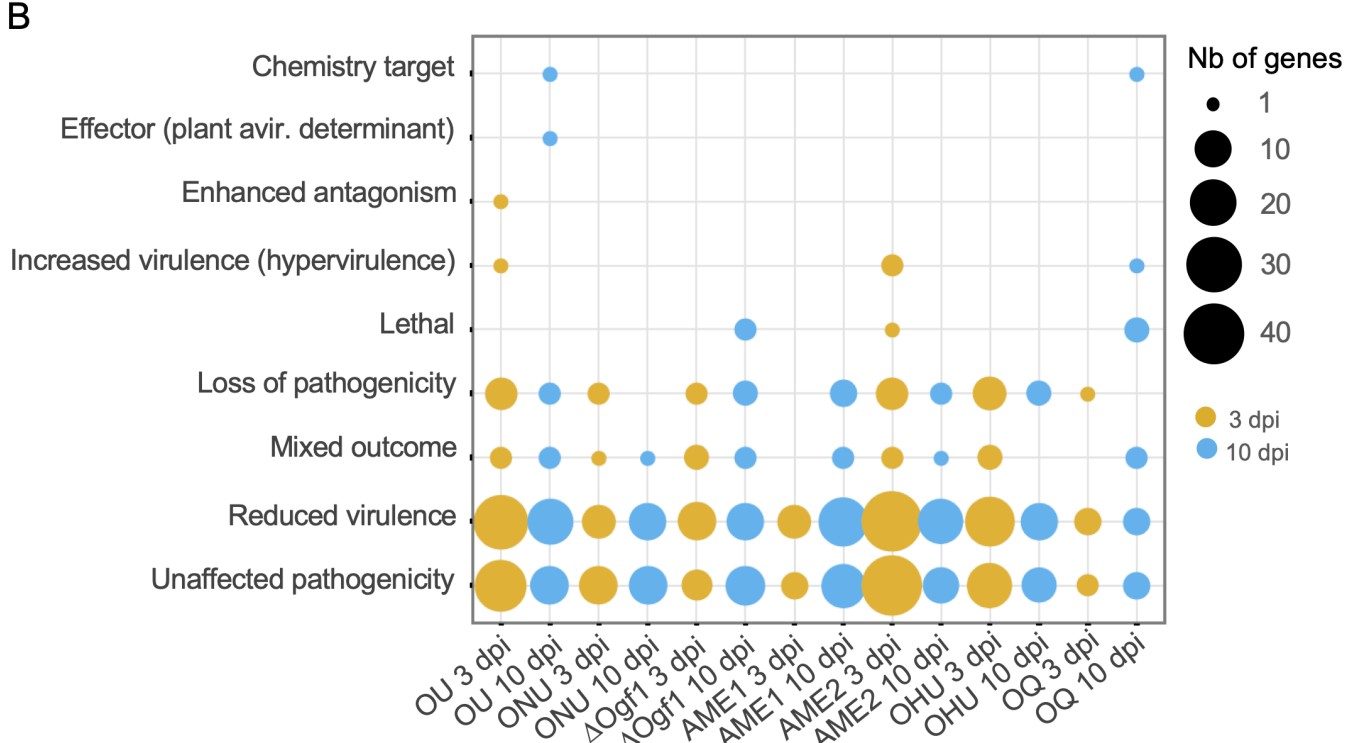

**FIG 6** Functional annotation analysis based on PHI-base of genes expressed at 3 and 10 dpi in *Ophiostoma* spp. strains (OU, ONU, ΔOgf1, AME1, AME2, OHU, and OQ) inoculated to *Ulmus americana*. (A) Principal component analysis of the predicted functional features in PHI-base. Each group is represented as the average position of all samples within the group and the standard errors of each PC as the error bars. (B) Number of representative DEGs with orthologs in PHI-base among *Ophiostoma* taxa and strains.

encoded unknown proteins including two probable apoplastic effectors (*OnuG4458* and *OnuG7061*) and one probable cytoplasmic effector (*OnuG7027*). The latter three genes were not DEG (Fig. 8).

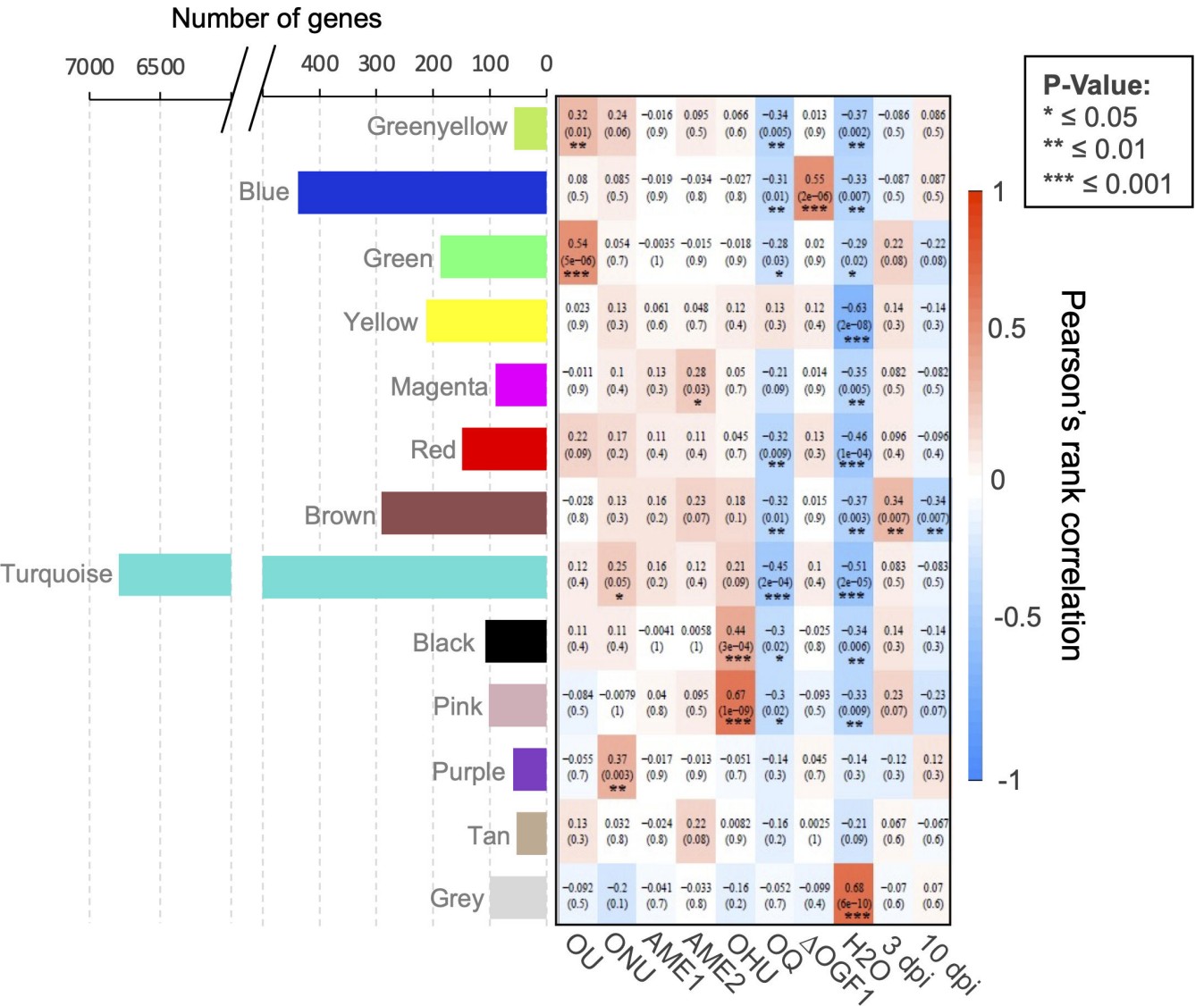

**FIG 7** Weighted gene correlation network analysis of genes expressed at 3 and 10 dpi in *Ophiostoma* spp. strains (OU, ONU, ΔOgf1, AME1, AME2, OHU, and OQ) inoculated to *Ulmus americana*. Module–trait relationships of module genes in different treatments. The number of genes is displayed in each module. Rows and columns correspond to module eigengenes and treatments (fungal strain or water control; overall fungal expression data at 3 and 10 dpi). Each cell indicates the Spearman's correlation coefficient (red color represents positive value and blue color represents negative value) and *P*-value between the module and gene expression.

The Brown module, which was associated with high gene expression at 3 dpi but low expression at 10 dpi, included 17 genes encoding secreted proteins. Within the latter, three genes coded for unknown proteins: *OnuG0138* (the hub gene in the Brown module), *OnuG7503*, and *OnuG8113*. Based on their molecular signatures, none of the unknown proteins is a probable effector. Two other potentially interesting genes in this module were *OnuG4204* and *OnuG2833* encoding tetraspanin and calcium channel subunit cch1, respectively (Fig. S4).

## DISCUSSION

Developments in genomics over the past two decades have improved our understanding of the DED fungi. For instance, nuclear genomes of nearly 200 strains of DED pathogens deposited in public databases (13, 44, 45) represent a highly valuable resource for comparative genomics. Transcriptomes of DED fungi based on expressed

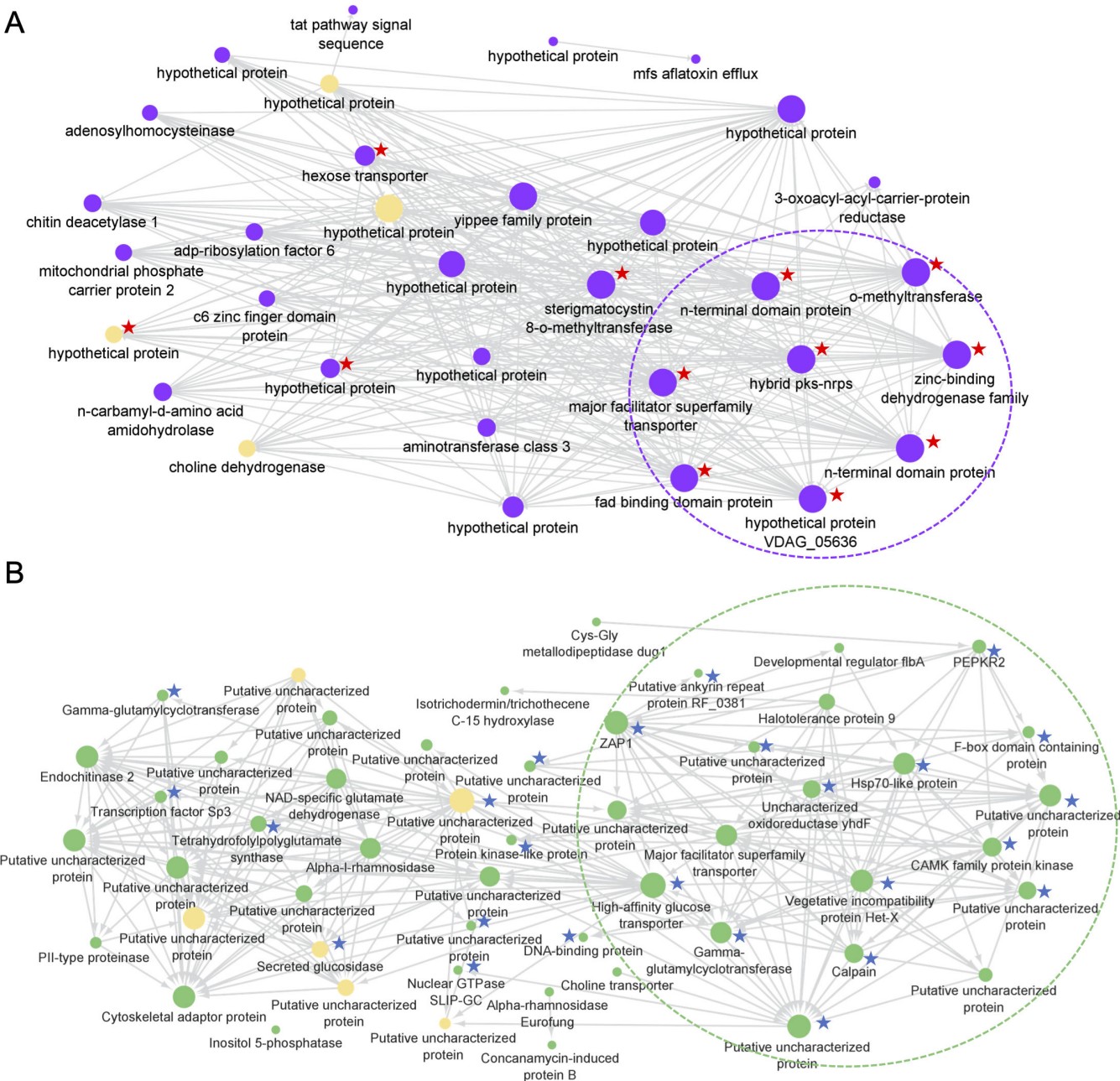

**FIG 8** Selected module-specific co-expression network analysis. Partial co-expression networks based on WGCNA analysis. The pale-yellow color represents genes encoding proteins with peptide signal and the node size is proportional to the number of links with other genes in the module. (A) Purple module: genes marked with a red star are DEGs at 10 dpi in ONU; the OPKS8 cluster is highlighted in the dotted region. (B) Green module: genes marked with a blue star are DEGs at 3 dpi in OU; the dotted region includes *zap1* (hub gene) and all genes with which it has a direct link (first neighborhood).

sequence tags provided expression data for roughly 25% of the predicted genes in *O. novo-ulmi* grown *in vitro* (46, 47), but this percentage fell to 1% for *Ophiostoma* transcriptomes recovered *in planta* (36). The development of the more efficient RNASeq technology (48) allowed more systematic, genome-wide investigations of gene expression in the DED fungi (49) both *in vitro* and *in planta*. The transition between yeast and mycelial growth (33, 34) was the first trait targeted for RNASeq studies in *O. novo-ulmi* subsp. *novo-ulmi*. Recently, Islam et al. (50) compared the genome-wide transcriptional responses of one sensitive and one resistant variety of *U. americana* in response to

inoculation with *O. novo-ulmi* subsp. *americana*. Fungal transcripts from the same *Ulmus-Ophiostoma* mRNA population were recovered in numbers sufficient for the analysis and identification of genes and pathways that may be relevant to the pathogenicity of DED fungi (38). Here, we continued our exploration of the *Ulmus-Ophiostoma* interaction and specifically compared the transcriptomic responses of different species and lineages of *Ophiostoma* species at "early" (3 dpi) and "advanced" (10 dpi) stages of development in the xylem of susceptible *U. americana* saplings.

Fungal reads accounted for a very small proportion (<0.25%) of reads recovered from inoculated elm saplings but were nevertheless twice more abundant than in previous transcriptomic studies of *U. americana-Ophiostoma* interactions (38). The fungal RNA fraction also included transcripts from endophytic fungi present in the stem of *U. americana* saplings used in the experiment (51, 52). Transcripts for homologs of genes *OnuG7913* (putative uncharacterized protein) and *OnuG7881* (AMP-dependent synthetase/ligase) were the most abundant in water-injected elms and accounted for 26% and 57% of the transcripts for the two genes in elms inoculated with either ONU or ΔOgf1. Conserved genes from endophytes made up the Gray module in the WGCNA statistical analysis (Fig. 7).

When the transcriptomes of all *Ophiostoma* strains were compared based on differential gene expression analysis at 3 and 10 dpi, fewer genes were detected in OQ. The latter also included a smaller subset of DEGs, especially at 3 dpi (Fig. 1A). This likely reflects that OQ is a saprotroph with very limited ability to colonize the xylem of healthy elms (26). Further transcriptomics studies of OQ grown on weakened trees or dead organic matter, such as sterilized wood blocks or elm sawdust extract agar (53), are needed to adequately describe the OQ transcriptomic landscape.

The genome of the highly virulent *O. novo-ulmi* subsp. *novo-ulmi* H327 was used as a reference for mapping and comparing transcriptomes of species, subspecies, and lineages investigated in this study. Although information may have been lost from the other species, the comparison with the mapping based on ONU allowed us to perform a deeper statistical analysis. The PCA analysis of the entire data set (excluding transcripts from the water-injected control) showed a clear distinction between pathogenic *Ophiostoma* species and the saprotroph OQ, whereas separation between the 3- and 10-dpi transcription results was not as pronounced (Fig. 2). Analyzing the more targeted KEGG, CAZy, and Phi-Base DEG data sets (Fig. 3–7) confirmed the separation between OQ and DED fungi and showed that, within the latter, variations in transcriptomes were not correlated with phylogenetic proximity. Thus, strains ONU and OHU were consistently close to each other, indicating highly similar gene expression profiles. In contrast, strains AME1 and AME2, which represent the *O. novo-ulmi* subsp. *americana* 1 and 2 lineages, were frequently further apart from each other than they were from other DED strains. This suggests that gene expression profiles are highly variable within subspecies *americana*. Deletion of gene *OnuG1537* encoding an ortholog of transcription factor GPF1 had a marked effect on ONU KEGG and CAZY DEG profiles (Fig. 4 and 5), as well as on the expression of several other genes (see below). This confirmed previous reports documenting multiple phenotypical changes, including loss of virulence, in Δ*gpf1* mutants in different fungi (39–41). Yet, in initial pathogenicity tests in which elm saplings were injected with $5 \times 10^4$ to $3 \times 10^5$ cells, ΔOgf1 was as virulent as WT strain H327. However, when elm inoculations were repeated using lower doses of inoculum ($1.25 \times 10^4$ or $3.12 \times 10^3$ cells), strain ΔOgf1 was found to be moderately, yet significantly less virulent than strain H327. This suggests a pathogen inoculum dose-dependent response of elm saplings, which we also observed in a null ONU mutant for gene *OnuG5955* encoding an alcohol oxidase (J. L. Sarmiento-Villamil and L. Bernier, unpublished data) as part of an ongoing re-evaluation of the virulence of *O. novo-ulmi* mutants from previous works (38).

The influence of incubation time within elm xylem was evident in the KEGG and PHI-Base DEG data sets, thus suggesting major changes in the metabolism of *Ophiostoma* spp. between 3 and 10 dpi. This is novel information since a single time point

(96 hpi) was considered in the previous transcriptomic analysis of elm-*Ophiostoma* interaction (38). Changes in gene expression were obviously linked to Carbohydrate Metabolism (Fig. 1B), as more than 80 genes were expressed at 3 dpi, whereas the number decreased to less than half at 10 dpi. Fungi that can grow on cellulose have access to an abundant and long-lasting resource. An organized growth pattern featuring extensively branching hyphae may aid in colonizing this substrate and creating the required high concentrations of extracellular enzymes to hydrolyze polysaccharides into soluble sugars, which can later be absorbed (54). The opposite trend was observed in the expression of genes linked to "electron transport." The "electron transport" chain is connected to fungal pathogenesis and is important for adaptation to challenges in the host environment, including nutrient limitation and the host immune response (55). Likewise, the number of genes linked to "metabolic processes" of "aromatic compounds" that were expressed at 10 dpi increased threefold compared to 3 dpi. Aromatic compounds released by plants are toxic to most fungi even at low levels, and therefore conversion of these compounds to less toxic metabolites is essential for fungi colonizing plants (56).

In functional annotation analysis based on Gene Ontology (Fig. 3), strains OU and AME2 stood out in numbers of DEGs at 3 dpi for the three categories of GO terms. The GO term with the highest number of genes was "hydrolase activity" in AME2 (35 DEGs at 3 dpi). Fungi produce a wide range of extracellular enzymes, in particular hydrolases, for degrading complex organic matter into smaller molecules, which they can then absorb (57). Fungal degradation of crystalline cellulose involves glycoside hydrolases that include endoglucanases, exoglucanases, and oxidative enzymes (58). First, endo-β-1,4-glucanases interact randomly to hydrolyze β-1,4-glycosides. This results in the production of terminals of the cellulose chain accessible to exoglucanases. DEGs in the GH6 and GH7 families were observed in OU, AME1, AME2, OHU, and OQ (Fig. 5B). Cellobiohydrolases in the GH6 and GH7 families attack the non-reducing and reducing ends, respectively, where they cleave cellobiose and cellotriose from the cellulose terminals (58–60). DEGs in family GH45 were detected in AME1 and ΔOgf1 at 3 dpi and in AME2 at 10 dpi. Family GH45 glycoside hydrolases are highly efficient, cellulose-specific endoglucanases (61, 62) used extensively in the textile industry for cellulase-based products (63).

While we did not look for evidence of cell wall erosion in elm xylem vessels, it is known that plant cell walls damaged by fungal cell wall degradation enzymes often release damage-associated molecular patterns that trigger plant immune responses. Cell surface-localized pattern recognition receptors perceive pathogen-associated molecular patterns (PAMPs) and initiate PAMP-triggered immunity, which involves a burst of reactive oxygen species, $Ca^{2+}$ influx, mitogen-activated protein kinase activation, and callose deposition. Callose formation blocks water transport to the leaves and, along with the obstruction of xylem vessels by fungal mycelium and spores and by plant-produced gels, gums, and tyloses, contributes to the wilting and eventual death of the host plant (3, 64–66).

Strains OU and AME2 once again stood out when DEGs with orthologs in PHI-base were analyzed, as they contained more DEGs than other strains (Fig. 6B). Furthermore, strain AME2 showed the strongest signal for upregulation early (3 dpi) in disease development, as shown by genes *OnuG1104* (expected to encode a member of GH31 with a signal peptide), *OnuG3630* (expected to encode a member of GH5 with a signal peptide), and *OnuG4204*, which encodes a tetraspanin. The different behavior of closely related strain AME1, both in terms of numbers and temporal occurrence of DEGs, was unexpected. Although the results suggest that upregulation of PHI-base DEGs (as well as KEGG DEGs) may account for the slightly lower virulence of AME1, pathogenicity tests including additional *O. novo-ulmi* subsp. *americana* strains and a higher number of replicates are required to validate this hypothesis.

The WGCNA identified 13 modules containing genes whose transcription profiles were associated with different *Ophiostoma* species and strains, or with different periods of incubation in elm tissue (Fig. 7). Two modules, Turquoise and Purple, were positively

correlated with ONU. The hub gene of module Turquoise (*OnuGp2723*) encodes profilin, a small actin-binding protein that is conserved among eukaryotic organisms (67) and has also been detected in viruses and cyanobacteria (68, 69). Although profilin is involved in many pathogenic interactions (67), it has never been associated with a fungal plant disease and this warrants further investigations.

Module Purple (Fig. 8A) contained all eight genes of the OpPKS8 cluster (*OnuG7305–OnuG7312*) encoding a putative fujikurin-like toxin (35). The first neighborhood connection between the OpPKS8 cluster and the hub gene *OnuG7303* suggests these physically close elements interact with each other. *OnuG7303*, along with *OnuG7306* from the OpPKS8 cluster, encode a putative sterigmatocystin 8-O-methyltransferase responsible for the biosynthesis of aflatoxin type B1 and B2 toxins (70). Gene *OnuG7303* was expressed when *O. novo-ulmi* subsp. *americana* MH75-4O colonized susceptible *U. americana* but not resistant (Valley Forge) elm (38). Should biochemical analyses confirm that DED fungi produce fujikurin- and aflatoxin-like toxins *in vitro* and *in planta*, the contribution of these molecules to virulence could be verified by functional analysis through the production of null mutants for genes *OnuG7303*, *OnuG7606,* and others to account for the genetic plasticity of fungal genomes (71).

The Blue module, positively correlated with ΔOgf1, was the second largest module and included several genes encoding transcription factors: *OnuG1809* (PHI: 1560), *OnuG3441*, *OnuG1220* (PHI: 1914), *OnuG5671*, *OnuG5928* (PHI: 1796), *OnuG3294* (PHI: 1710), *OnuG0079* (PHI: 1354), and *OnuG5077.1*. As transcripts for these genes were more abundant in strain ΔOgf1 than in other strains (Table S3), this implies that the deletion of *ogf1* possibly stimulated the upregulation of other transcription factors whose role in the pathogenicity of *O. novo-ulmi* should be investigated by knockout of the genes that encode them.

In the Green module positively correlated with the moderately aggressive *O. ulmi*, the hub gene *(OnuGp6492)* was an ortholog of *zap1* encoding a zinc-responsive transcriptional regulator protein (Fig. 8B). *zap1* controls zinc uptake and regulates the transcription of the *zrt1* and *zrt2* genes in response to zinc availability in *Saccharomyces cerevisiae* (72). The regulation of zinc acquisition by *zap1* is fundamental for fungal pathogenesis in mammalian hosts (73). For instance, *zap1* regulates zinc homeostasis and modulates virulence in *Cryptococcus gattii*, one of the causal agents of cryptococcosis (74). *zap1* was DEG in OU at 3 dpi (log$_2$FC = 3.91), whereas *zrt1* was DEG in OU at 10 dpi (log$_2$FC = −4.27). In ONU, however, the log$_2$FC values for *zap1*, *zrt1,* and *zrt2* were low (0.48, 0.001, and 0.54, respectively). This suggests that the investigation of a possible link between zinc acquisition and virulence in DED fungi should include the production and analysis of null mutants in both *O. ulmi* and *O. novo-ulmi*.

Furthermore, WGCNA showed that *zap1* was directly linked to several other genes (Fig. 8B). These included *OnuG4429* encoding a calpain (calcium-activated neutral proteinase), a protein involved in fungal development and pathogenicity (75, 76). *zap1* was also directly linked to *OnuG2192* encoding an Hsp70-like protein, which, in *Cryptococcus neoformans*, influences the interaction between the pathogen and its host (75). Hsp70 may also play a dual role during infection since it can act as an effector molecule inducing nitric oxide (NO) production by epithelial cells or, conversely, as an immunosuppressive molecule reducing NO production by macrophages (76, 77). In plants, the inhibition of NO synthesis compromises the hypersensitive disease resistance response of *Arabidopsis* leaves to *Pseudomonas syringae* infection, thereby promoting disease and bacterial growth (78). The role of NO was also shown in soybean cells where reactive oxygen intermediates induce genes for the synthesis of protective natural products (79). In the present work, *zap1* and *hsp70*-like genes were upregulated (log$_2$FC = 3.91 and 3.35, respectively) only in the moderately virulent *O. ulmi* at 3 dpi, suggesting they play a role as virulence modulators during the earlier phase of disease development.

The Greenyellow module, which was also positively correlated with OU, included four genes that were DEG at 10 dpi: *OnuG1485* (PHI: 112), *OnuG3416*, *OnuG6502,* and *OnuG6506* (PHI: 438). *OnuG1485* encodes a monooxygenase, whereas the predicted

product of *OnuG6506* is a pisatin demethylase (PDA). Pisatin is a plant phytoalexin produced in response to microbial attack. However, pisatin can be biotransformed by PDA into a molecule less harmful to fungal pathogens (80).

The Black and Pink modules were positively correlated with OHU and contained only genes that were upregulated at 3 dpi (Table S6). Four genes in the pink module have orthologs in PHI-base, including *OnuG1398* (PHI: 2022), *OnuG3407* (PHI: 58), and *OnuG3720* (PHI: 2022), which are involved in the synthesis of melanin, an amorphous polymer that increases the ability of fungi to survive in a hostile environment (81) and is a virulence factor in pathogenic fungi (82, 83). The fourth gene, *OnuG1395* (PHI: 2921), is linked to iron absorption. The deletion of the ortholog of this gene in *Colletotrichum graminicola* resulted in a reduction in virulence (84). Additional *Ophiostoma* genes of interest with orthologs in the PHI-base might be present in the Magenta module associated with AME2. For instance, deletion or disruption of orthologs of *OnuG1084* (PHI:423), *OnuG366* (PHI:2020), *OnuG5175* (PHI:504), *OnuG7194* (PHI:881), or *OnuG4969* (PHI:2292) led to reduced virulence in various species (85–89), whereas inactivation of the ortholog of *OnuG5266* (PHI:2177) in *M. oryzae* enhanced conidiation (90).

Gene clustering by WGCNA also highlighted genes whose expression differed markedly according to time. Thus*, OnuG4204*, one of two genes predicted to encode tetraspanins in ONU (32), was the hub gene in the Brown module that was positively correlated with global expression at 3 dpi (Fig. 7; Fig. S4). Tetraspanins are membrane proteins that act as organizers of membrane-signaling complexes required for the pathogenicity of plant pathogenic ascomycetes (91). *OnuG4204* had a direct connection with another hub gene, *OnuG*2833, which codes for calcium channel subunit cch1, the only high-affinity calcium channel in the plasma membrane of fungal cells (85). Expression of this gene is linked to cell stress signaling (92). *OnuG4204* and *OnuG*2833 were both DEGs at 3 dpi in strain OHU ($log_2FC$ = 2.30 and 2.64, respectively), thereby suggesting concerted activation in these genes in OHU.

A comparison of the 50 genes with the most variable expression in OU and ONU (Fig. S2) showed that several genes encoding relatively small unknown proteins were upregulated in the less virulent species *O. ulmi* either early (*OnuG*0653, *OnuG*3998, *OnuG*8505, and *OnuG*8458; Green module) or late (*OnuG*5230; Turquoise module) in disease development. These DEGs were found almost only in OU except for *OnuG*8505 (DEG at 10 dpi in AME2 with $Log_2FC$ = −2.43) and *OnuG5230* (DEG at 10 dpi in AME1 with $Log_2FC$ = −2.05, AME2 with $Log_2FC$ = −2.88, and OHU with $Log_2FC$ = −2.77). Although the predicted products of these genes lack a signal peptide, they may nevertheless represent effectors secreted by non-conventional processes (86) and, therefore, their role in pathogenicity should be investigated.

## Conclusions

This work compared for the first time the transcriptomes of different *Ophiostoma* taxa inoculated to American elm. Overall, we observed taxon-specific transcriptome profiles, with transcriptomes of the saprotroph *O. quercus* significantly different from transcriptomes of DED pathogens. For the pathogenic taxa analyzed, the transcriptome variations were not associated with phylogenetic relationships. Gene network analyses showed the diversity and complexity of gene interactions among strains, which is further exemplified by a significant rewiring of gene interactions by the deletion of the gene encoding a homolog of the GPF1 transcription factor. In fact, a mutation in *O. novo-ulmi* subsp. *novo-ulmi* at this single locus silenced the gene encoding cerato-ulmin, downregulated several genes in the OpKS8 cluster, and upregulated a subset of genes encoding other transcription factors. This resulted in a moderate, yet significant decrease in virulence. Our study provides a spectrum of candidate pathogenicity genes in the DED fungi, which should be further examined by functional genetic studies.

## MATERIALS AND METHODS

### Plant material and growth conditions

On the 22 January 2020, dormant, 2-year-old saplings of *Ulmus americana* grown from seeds collected on the Laval University campus (Québec City) were taken indoors, allowed to thaw for 1 day, and transferred to a growth chamber kept at 10°C without light. Over the next 2 weeks, growth conditions were progressively brought to the following conditions: 16 h light at 24°C (±2) and 8 h darkness at 18°C (±2), with 60% relative humidity and light intensity at 725 lux. Saplings were inoculated 28–30 days after bud break once leaves were fully expanded.

### Inoculation procedure and sampling of elms

Prior to inoculation, seven strains of *Ophiostoma* spp. were recovered from −80°C storage at the Centre d'Étude de la Forêt collection (http://www.cef-cfr.ca/index.php?n=CEF.Collections). Strains included the moderately aggressive *O. ulmi* Q412T-O (OU), the highly aggressive *O. novo-ulmi* subsp. *novo-ulmi* H327-O (ONU), *O. novo-ulmi* subsp. *americana* lineage 1 DDS100 (AME1), *O. novo-ulmi* subsp. *americana* lineage 2 DDS154 (AME2), *O. himal-ulmi* HP30 (OHU), the saprobe *O. quercus* AU5-1 (OQ), as well as *O. novo-ulmi* subsp. *novo-ulmi* Δ13–15 (ΔOgf1) in which gene *OnuG1537* encoding a homolog of transcription factor GPF1 had been deleted by targeted mutagenesis of the highly aggressive strain H327-OΔmus52 using the procedure developed by Sarmiento-Villamil et al. (87). Yeast cells were obtained by incubating strains in liquid *Ophiostoma* minimal medium (88) on a rotary shaker (130 rpm) at 21°C for 4 days. Fungal cultures were filtered through eight layers of cheesecloth and centrifuged for 5 min at 5,000 rpm, and yeast cells were resuspended in sterile distilled water at a concentration of $4 \times 10^6$ cells mL$^{-1}$.

Saplings used in the transcriptomic analysis of the elm-*Ophiostoma* interactome were inoculated on 20 March 2020 by injecting 25 µL of *Ophiostoma* yeast cell suspension into each of three holes (total inoculum = $3 \times 10^4$ cells) drilled with a 3/32 bit (89) at ca. 20 cm up the main stem. Non-inoculated control saplings were injected with sterile distilled water. Inoculation/injection holes were covered with Parafilm (Bemis Co., Neenah, WI, USA). Each treatment was applied to four saplings distributed randomly within four different blocks. Each block also included one sapling that received no treatment, for a total of 68 saplings. Because of crown size expansion between bud break and inoculation, saplings had to be incubated in two growth chambers set according to the parameters described above. Block 1 was placed in one growth chamber, whereas blocks 2–4 were placed in the other growth chamber. Stem samples for transcriptomic analyses were collected at 0-, 3-, and 10-days post-inoculation. External foliar symptoms (wilting, yellowing, or browning of leaves) were noted at 0, 3, and 10 dpi, and treatments were rated on a scale from 0 (no symptomatic leaves) to 5 (>90% symptomatic leaves) (Table S1.1). Saplings that had been treated with *Ophiostoma* or distilled water were sampled ca. 2 cm above the section of the stem that had been inoculated or injected. The stem of untreated (0 dpi) saplings was sampled at a similar height. For each time point, four destructive biological replicates were generated. Stem samples were quickly frozen in liquid nitrogen and stored at −80°C.

The virulence of mutant ΔOgf1 and its H327Δmus52 parental strain was also assessed on Golden Delicious apples and three additional sets of *U. americana* saplings. Apples (eight biological replicates per treatment) were inoculated with plugs of mycelium and the diameter of necroses was measured at 14 dpi, as previously described (93). The first set of inoculated saplings (20 March 2020, eight biological replicates per treatment, $1 \times 10^5$ cells inoculated per sapling) was placed in the same growth chamber as block 1 saplings used in the transcriptomics study. Development of external foliar symptoms was monitored over 14 days, and treatments were rated on a scale from 0 (no symptoms) to 4 (>75% symptomatic leaves) (Table S1.2). The second and third sets of saplings were inoculated on 25 June 2020 and 16 June 2021, respectively, as part of larger-scale testing of *O. novo-ulmi* insertional mutants. In the 2020 trial (Table S1.3), saplings (10

biological replicates per treatment) were inoculated with $5 \times 10^4$ cells, whereas saplings (eight biological replicates per treatment) inoculated in 2021 (Table S1.4A) received lower doses of inoculum ($1.25 \times 10^3$ or $3.12 \times 10^3$ cells). Inoculated and water-injected controls were maintained in a greenhouse compartment as previously described (87) and external symptoms were assessed over 21 days (2020) or 35 days (2021). Percent symptomatic leaves (including leaves that had fallen off) were recorded for each sapling, and the mean proportion of symptomatic leaves was calculated for each treatment. Development of *O. novo-ulmi* in the xylem of saplings inoculated in 2021 (Table S1.4B) was evaluated by reisolating from tissue around the inoculation point, at mid-stem, and in the distal portion of the stem. Strains of *O. novo-ulmi* were identified morphologically among cultures that developed on a semi-selective medium containing cycloheximide, streptomycin sulfate, and chloramphenicol (53, 94).

## RNA extraction and sequencing

Stem samples were ground with a Mixer mill 400, set at 30 seconds, frequency 30. The entire grinding process was carried out in the presence of liquid nitrogen, and ground samples were immediately stored at −80°C. Total RNA was extracted by the cetrimonium bromide (CTAB) protocol (92) and quantified in a Thermo Scientific Nanodrop 1000 Spectrophotometer (RRID: SCR_016517). The integrity of RNA extractions was checked in a Bioanalyzer RNA 6000 Nano assay. All samples were diluted to 35 ng µL$^{-1}$ in a final volume of 30 µL, sent to Centre d'Expertise et de Services, Génome Québec (Montréal, QC, Canada) and sequenced on the Illumina HiSeq 1000 platform.

## Sequence data processing, mapping, and annotation

Raw sequence data quality was visualized using FastQC v.0.11.9 (95). All samples were filtered with Trimmomatic v.0.36 (paired-end mode; minimum length 100 pb) (96). For the analysis of fungal transcriptomes, all samples were mapped and aligned onto the *O. novo-ulmi* H327 genome (32) with STAR read mapper v.2.7.8a (97) using default parameters. Mapped reads were annotated according to the *O. novo-ulmi* H327 functional annotation (32) based on KEGG (98), GO (99), CAZy (100), and PHI-base (101). Differentially expressed genes were selected when comparing transcript levels at 3 and 10 dpi of each fungal organism (or water control). Gene Ontology term enrichment analyses were performed on sets of overexpressed fungal genes at 3 and 10 dpi as described previously (33).

## Statistical analyses

All statistical analyses were performed using R Studio v.1.4.1106. To enable the representation of differential expression analyses between samples, all read counts were normalized with DESeq2 package in R (102) with a fold change of two ($|\log_2 FC| > 2$) and FDR < 0.05 using scatter plot visualization as previously described (103). Variability among fungal transcriptomes was assessed by PCA and computed using the *rda()* function in R from *Vegan* package. Heatmap of clustered results was based on normalized and transformed read counts with the *vst()* function in DESeq2 package in R. Venn diagrams were produced with the Bioinformatics & Evolutionary Genomics platform (http://bioinformatics.psb.ugent.be/webtools/Venn/). Gene transcription patterns in the different *Ophiostoma* species, lineages, and genotypes were also investigated by WGCNA using the *WGCNA* package from R (42). The gene matrix was constructed with a threshold of 12. All modules were hierarchically clustered based on topological overlap matrix similarity. Interaction networks within selected gene modules were identified using Cytoscape v3.9.0 (104). Statistical analyses of all inoculations in elms and Golden Delicious apples with *Ophiostoma* spp. were performed using Past4 v.4.03 (Table S1).

## ACKNOWLEDGMENTS

The authors thank Isabelle Giguère, André Gagné, and Jean-Guy Catford for technical support.

This research was funded by Genome Canada, Genome British Columbia, and Génome Québec within the framework of project bioSAFE (Biosurveillance of Alien Forest Enemies, Project number 10106), and by The Natural Sciences and Engineering Research Council of Canada (NSERC Discovery Grant RGPIN-2018-06607).

T.C.O. and L.B. conceived and designed the experiments. T.C.O., N.J.F., I.P., P.T., and L.B. performed transcriptional analysis curation and comparative analyses. T.C.O. and N.J.F. performed bioinformatics analyses. T.C.O., J.L.S.V., I.P., P.T., and L.B. contributed reagents, materials, and analysis tools. T.C.O. and L.B. wrote the manuscript. All authors read and approved the final manuscript.

## AUTHOR AFFILIATIONS

[1]Institut de Biologie Intégrative et des Systèmes, Université Laval, Québec, Quebec, Canada

[2]Centre d'étude de la Forêt, Faculté de foresterie, de géographie et de géomatique, Université Laval, Québec, Quebec, Canada

[3]Department of Natural Resource Sciences, McGill University, St. Anne-de-Bellevue, Quebec, Quebec, Canada

[4]Instituto de Hortofruticultura Subtropical y Mediterránea, Consejo Superior de Investigaciones Científicas-Universidad de Málaga (IHSM-CSIC-UMA), Estación Experimental "La Mayora", Málaga, Spain

[5]Canadian Forest Service, Natural Resources Canada, Laurentian Forestry Centre, Québec, Quebec, Canada

## AUTHOR ORCIDs

Thais C. de Oliveira  http://orcid.org/0000-0002-7565-7563
Louis Bernier  http://orcid.org/0000-0002-1789-8190

## FUNDING

| Funder | Grant(s) | Author(s) |
|---|---|---|
| Genome Canada (GC) | 10106 | Ilga Porth |
| | | Louis Bernier |
| | | Philippe Tanguay |
| Genome British Columbia (Genome BC) | 10106 | Ilga Porth |
| | | Louis Bernier |
| | | Philippe Tanguay |
| Génome Québec (GQ) | 10106 | Ilga Porth |
| | | Louis Bernier |
| | | Philippe Tanguay |
| Gouvernement du Canada | Natural Sciences and Engineering Research Council of Canada (NSERC) | RGPIN-2018-06607 | Louis Bernier |

## AUTHOR CONTRIBUTIONS

Thais C. de Oliveira, Conceptualization, Data curation, Formal analysis, Investigation, Methodology, Project administration, Software, Validation, Visualization, Writing – original draft, Writing – review and editing | Nastasia J. Freyria, Data curation, Formal analysis, Investigation, Methodology, Software, Visualization, Writing – review and editing | Jorge Luis Sarmiento-Villamil, Conceptualization, Data curation, Formal analysis, Methodology, Writing – review and editing | Philippe Tanguay, Conceptualization, Data

curation, Funding acquisition, Investigation, Writing – review and editing | Louis Bernier, Conceptualization, Data curation, Formal analysis, Funding acquisition, Methodology, Resources, Supervision, Validation, Writing – review and editing.

## DATA AVAILABILITY

The transcript sequences are available in GenBank under the accession number PRJNA856292, ID 856292.

## ADDITIONAL FILES

The following material is available online.

## Supplemental Material

**Supplemental Figures S1 to S4 (Spectrum03694-23-S0001.pdf).** Supplementary figures.
**Supplemental Tables S1 to S6 (Spectrum03694-23-S0002.xlsx).** Supplemental tables.

## Open Peer Review

**PEER REVIEW HISTORY (review-history.pdf).** An accounting of the reviewer comments and feedback.

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
