## [Reviewer comments · Microbiology Spectrum]

Microbiology Spectrum

Unraveling the transcriptional features and gene expression networks of pathogenic and saprotrophic *Ophiostoma* species during the infection of *Ulmus americana*

Thais de Oliveira, Nastasia Freyria, Jorge Sarmiento-Villamil, Ilga Porth, Philippe Tanguay, and Louis Bernier

Corresponding Author(s): Thais de Oliveira, Universite Laval

Review Timeline:

Submission Date:	October 17, 2023
Editorial Decision:	November 15, 2023
Revision Received:	December 7, 2023
Accepted:	December 8, 2023

Editor: Lindsey Burbank

Reviewer(s): The reviewers have opted to remain anonymous.

Transaction Report:

DOI: <https://doi.org/10.1128/spectrum.03694-23>

Re: Spectrum03694-23 (Unraveling the transcriptional features and gene expression networks of pathogenic and saprotrophic *Ophiostoma* species during the infection of *Ulmus americana*)

Dear Miss Thais Campos de Oliveira:

Thank you for the privilege of reviewing your work. Below you will find my comments, instructions from the Spectrum editorial office, and the reviewer comments.

The general response is very positive, your study provides significant information to better understand an important pathosystem. The reviewers have made a number of suggestions for better clarity of your manuscript and easier understanding by the reader. Please address these suggestions as outlined below.

Revision Guidelines

Sincerely,
Lindsey Burbank
Editor
Microbiology Spectrum

Reviewer #1 (Comments for the Author):

This study reports on important advancements in our understanding of the Dutch elm disease pathosystem. The authors applied cutting-edge molecular techniques and thorough analyses that provided valuable insights into the interactions between tree host and associated microbes, both pathogenic and saprophytic. Most importantly, this study does not rely solely on information

gleaned from genome content, but rather highlights the importance of analyzing gene expression during the infection process in planta. These analyses identified candidate genes associated with pathogenicity in *Ophiostoma ulmi* and *O. novo-ulmi* and produced a hypovirulent knockout mutant, demonstrating the importance of the hydrophobin cerato-ulmin as a factor contributing to virulence. These findings have major implications for management of DED.

The paper is very well organized and well written. My suggestions for edits are generally intended to improve clarity for the reader.

Since the Materials and Methods section is located at the end of the manuscript, I think it will be necessary to place any abbreviations described in this section to the beginning of the manuscript. Especially OU, ONU, OQ, etc. since these are used extensively throughout the Results and Discussion sections earlier in the manuscript.

L107: Can you elaborate more on this or state it in a more descriptive manner? What specifically is "lagging behind" and what is it lagging behind? Do you mean to say that there are relatively fewer studies on this pathosystems vs. others? I think this statement is important for this manuscript but needs more context.

L123-124: I would suggest using more descriptive terms to describe the symptoms expressed under various experimental treatments and at different time points. What are the "symptoms of wilting" that are measured or otherwise observed? I'm assuming there was no measurement of extent of colonization in the xylem. Is it only the leaf wilting that was observed? How did the symptoms expressed at 3 days differ from the trees that were inoculated with same/different strains or species at 10 days. Here you said that leaf wilting was observed at 3 dpi and that "intermediate symptoms" were observed at 10 dpi. Are these the same levels of symptoms? Were the "intermediate" symptoms at 10 dpi more advanced than the wilting at 3 dpi?

L130: How do the data related to "virulence" of isolates differ between inoculated saplings vs. inoculated Golden Delicious apples? The "symptoms" that are measured related to virulence are very different in these two systems. If using the term "virulence" for both types of inoculations, please describe how they differ. It sounds like "virulence" on apples is related to size of necrotic spots surrounding the point of inoculation, whereas those on sapling are related to wilting. Do you think these metrics are equivalent in terms of virulence? In other words, is an isolate that produces larger necrotic spots on apples expected to produce more severe symptoms in inoculated seedlings? Has there been any evidence that these are related?

L151: This is one of only two instances where cpm is used in the manuscript. Please include a description for this. It is probably not common knowledge for most readers.

L159: For DEG analyses, did you only compare 3dpi and 10dpi? I think it would have been interesting to see how gene expression in planta compared with *Ophiostoma* isolates in vitro. This would have given an opportunity to compare "baseline" or background gene expression levels for the genes of interest when the fungus is not interacting with a live plant host. Maybe these genes are only expressed when interacting with a live host? Do we know this to be true?

L161: It looks like this is the first use of the acronym DEG in the manuscript in its current order. If this is the order that it will be when published, I would suggest describing DEG here for the reader.

LL209: I would suggest rewording this sentence slightly to clarify that the gene expression of strain AME2 was more like that of OU than AME1. This is somewhat vague as it is currently written. Rewording should make a clearer distinction between similar gene expression vs. close phylogenetic relationship.

L233-234: Similar to my comment above, please clarify the first statement in this paragraph by saying something like "...orthologs in the CAZy database confirmed that gene expression in strain OQ strongly differed..." To make it clear that you are talking about differences in gene expression and not other characteristics.

L258: For those not familiar with WGCNA, it is not clear what the various colors/modules represent. I would suggest that you include a brief introductory sentence to this paragraph with some information so that the reader can better interpret your results.

L326-328: When it is discovered that some genes expressed in trees inoculated with *Ophiostoma* spp. are also present in elms that were mock-inoculated with water, is it possible to remove these from the analysis since they are unlikely to be attributable to the pathogens? Maybe you already did this and its not in the paper? Or maybe I missed it? It would be interesting to discuss here.

L345-350: Similar to my earlier comments regarding differences in gene expression among taxa- please include some terms in this paragraph to clarify that separation among these groups in the PCA plots reflects differences in gene expression. What does it mean for these groups to be closer together or further apart in the PCA plot? I know this seems obvious, but it will be easier for the reader to interpret if you include some context. Here is an example with some suggested edits: "Thus, strains ONU and OHU were consistently close to each other, indicating similar gene expression profiles. whereas strains AME1 and AME2 representing *O. novo-ulmi* ssp. *americana* lineages 1 and 2 were often further apart from each other than they were from other DED strains, suggesting that gene expression profiles within this lineage are highly variable.

L357-359: It might be good to provide some speculation as to why the deletion of this gene had this effect. It sounds like ceratoulmin has an important function as hydrophobin, which serves several biological roles according to the literature. Is there any biological reason why you think removing this gene would diminish pathogenicity? Any speculation as to why there is a dose-dependent effect when inoculating with this strain?

L364: This change in gene expression in genes related to carbohydrate metabolism seems significant. Can you provide some interpretation in a biological context? Any reason these would be upregulated at 10 dpi and not 3 dpi or 96 hpi? What biological functions of *Ophiostoma* spp. would rely on these gene products during this stage of the infection process? If there is any information on this in the existing literature, please cite.

L387-396: Was there any upregulation (or downregulation?) of genes related to CWDEs? Is there any evidence that *Ophiostoma* spp. produce these enzymes? Is cell wall degradation known to be important for *Ophiostoma* pathogenicity? I do not generally think of these organisms as causing any form of necrosis or enzymatic cell wall degradation, but maybe there is evidence for this elsewhere in the literature?

L413: Very interesting! Seems significant.

L428-430: It is not immediately clear how these results support the conclusion of "compensation". Can you provide more context here for the reader? If you have confirmed that the mutant Δogf1 does not produce the gene product, then what are these other transcription factors compensating for?

L562: Which external symptoms were assessed specifically? Were there any measurements taken or was it purely qualitative? Please provide more information here.

Figure 7: If the numbers in the red, white, and blue boxes are meant to be interpreted by the reader, please increase the font size.

Figure 8: Same as above. If text is meant to be interpreted by the reader, suggest increasing font size.

Reviewer #2 (Comments for the Author):

The manuscript untitled "Unraveling the transcriptional features and gene expression networks of pathogenic and saprotrophic *Ophiostoma* species during the infection of *Ulmus americana*" (control no. Spectrum03694-23) by de Oliveira and co-workers describes a transcriptomic comparative analysis of seven fungal plant pathogen strains, *Ophiostoma* sp., responsible for Dutch elm disease. These strains include mutant strain and a saprotroph strain with low to no virulence. The results of this study are novel and interesting for the field due to the -in planta- setting of the experiment. In planta transcriptomic analysis is known to be more difficult due to complexity of the samples and resulting background noise. This study shows very good quality data validated by statistical tests.

The transcriptomic analysis presented here has been done at two time points: 3 days and 10 days after infection. As expected with this kind of analysis, a large number of genes show a differential expression among strains and conditions. This work is a good basis for further gene candidate functional characterization.

Page lines 125-130. Inoculum concentration on virulence, this paragraph is not clear to me. What was the purpose of testing several inoculum concentration? Is it to see the immune response? And, how many plants species were tested and why?

Page 9 line 184. OpKS8 putative gene cluster, what are the annotated function of these genes? Based on fig. S2, gene 7303 is also upregulated, is it part of the same OpKS8 gene cluster?

Page 11, line 232, CAZyme-encoding genes. The Auxiliary Activity family AA (lignin/aromatic active enzymes) is not represented in figure 5. Are they missing from the analysis?

What about the known regulators in other species such as Carbon repressor, cellulases, hemicellulases and pectin regulators?

A comment on the format, at first, I found difficult to understand and follow the results due to the use of abbreviations such as pdi, ONU, OU, AME1 etc. DED was defined in the introduction, but all the other abbreviations were only mentioned in the mat and met section (at the end of the manuscript). I understand it is a format issue but to ease the reading I suggest simply defining them in the introduction: example line 70-71, *Ophiostoma ulmi* (OU).

There are a few spelling mistakes to check throughout the suppl. files both table titles and figures + text. (example table S4, hbighly).

Unraveling the transcriptional features and gene expression networks of pathogenic and saprotrophic *Ophiostoma* species during the infection of *Ulmus americana*

Thais C. de Oliveira^{1,2*}, Nastasia J. Freyria³, Jorge Luis Sarmiento-Villamil^{1,2,4}, Ilga Porth^{1,2}, Philippe Tanguay⁵ and Louis Bernier^{1,2*}

¹ Institut de Biologie Intégrative et des Systèmes, Université Laval, Québec, QC, Canada

² Centre d'étude de la Forêt, Faculté de foresterie, de géographie et de géomatique, Université Laval, Québec, Canada

³ Department of Natural Resource Sciences, McGill University, St. Anne-de-Bellevue, QC, Canada

⁴ Instituto de Hortofruticultura Subtropical y Mediterránea, Consejo Superior de Investigaciones Científicas-Universidad de Málaga (IHSM-CSIC-UMA), Estación Experimental "La Mayora", Avda. Dr. Wienberg, s/n, E-29750 Algarrobo-Costa, Málaga, Spain

⁵ Canadian Forest Service, Natural Resources Canada, Laurentian Forestry Centre, Québec, QC, Canada

We thank the editor and reviewers for their time and comments, which helped us improve the quality of our manuscript. We have addressed all comments and included explanations for modifications we made. We believe that the changes and additions we have made to our manuscript (highlighted in yellow in the copy with track changes) have helped strengthen our conclusions and highlight the importance of this work.

Reviewer #1 (Comments for the Author):

This study reports on important advancements in our understanding of the Dutch elm disease pathosystem. The authors applied cutting-edge molecular techniques and thorough analyses that provided valuable insights into the interactions between tree host and associated microbes, both pathogenic and saprophytic. Most importantly, this study does not rely solely on information gleaned from genome content, but rather highlights the importance of analyzing gene expression during the infection process in planta. These analyses identified candidate genes associated with pathogenicity in *Ophiostoma ulmi* and *O. novo-ulmi* and produced a hypovirulent knockout mutant, demonstrating the importance of the hydrophobin cerato-ulmin as a factor contributing to virulence. These findings have major implications for management of DED.

The paper is very well organized and well written. My suggestions for edits are generally intended to improve clarity for the reader.

Since the Materials and Methods section is located at the end of the manuscript, I think it will be necessary to place any abbreviations described in this section to the beginning of the

manuscript. Especially OU, ONU, OQ, etc. since these are used extensively throughout the Results and Discussion sections earlier in the manuscript.

We thank the reviewer for highlighting this mistake. We included abbreviations for each organism in the introduction of the manuscript.

L107: Can you elaborate more on this or state it in a more descriptive manner? What specifically is "lagging behind" and what is it lagging behind? Do you mean to say that there are relatively fewer studies on this pathosystems vs. others? I think this statement is important for this manuscript but needs more context.

We meant to say that published omics studies of DED pathogens were conducted mostly *in vitro* rather than *in planta*. The sentence has been modified (L103) and we hope it is now clearer to readers.

L123-124: I would suggest using more descriptive terms to describe the symptoms expressed under various experimental treatments and at different time points. What are the "symptoms of wilting" that are measured or otherwise observed? I'm assuming there was no measurement of extent of colonization in the xylem. Is it only the leaf wilting that was observed? How did the symptoms expressed at 3 days differ from the trees that were inoculated with same/different strains or species at 10 days. Here you said that leaf wilting was observed at 3 dpi and that "intermediate symptoms" were observed at 10 dpi. Are these the same levels of symptoms? Were the "intermediate" symptoms at 10 dpi more advanced than the wilting at 3 dpi?

When assessing phenotypic changes in elms following treatments, we only monitored external symptoms. The earliest symptoms that we observed were wilting of leaves (within 3 dpi), whereas yellowing, browning and abscission of leaves occurred later during the interaction. The extent of symptoms on elms inoculated in March 2020 for transcriptomics (Table S1.1) and comparative host response to strains ONU and *Δogf1* (Table S1.2) was estimated on scales of 0 to 5 and 0 to 4, respectively. In the case of elms inoculated in June 2020 (Table S1.3) or June 2021 (Table S1.4), percent symptomatic leaves (including leaves that had fallen off) was recorded for each sapling and the mean proportion of symptomatic leaves was calculated for each treatment. We included additional information on symptom assessment in subsection "*Inoculation procedure and sampling of elms*" of the "Materials and Methods" section (L546-549; L558-560; L567-569). We also modified subsection "*External symptoms in American elm saplings inoculated with *Ophiostoma spp.**" in "Results" in order to clarify our observations (L120-124).

L130: How do the data related to "virulence" of isolates differ between inoculated saplings vs. inoculated Golden Delicious apples? The "symptoms" that are measured related to virulence are very different in these two systems. If using the term "virulence" for both types of inoculations, please describe how they differ. It sounds like "virulence" on apples is related to size of necrotic spots surrounding the point of inoculation, whereas those on sapling are related to wilting. Do

you think these metrics are equivalent in terms of virulence? In other words, is an isolate that produces larger necrotic spots on apples expected to produce more severe symptoms in inoculated seedlings? Has there been any evidence that these are related?

The reviewer is right: virulence on elm saplings is related to foliar symptoms (including wilting), whereas virulence on apples is related to the size (diameter) of necrotic lesions. This is now clarified in L128-129 and L555 of the revised manuscript. Although these symptoms are very different, Plourde and Bernier (2014) showed that apple necroses at 28 dpi and *Ulmus procera* defoliation were positively correlated when *O. novo-ulmi* ssp. *novo-ulmi* strains were inoculated to these hosts. We use the GD apple assay for phenotyping wild-type and mutant strains of *O. ulmi* and *O. novo-ulmi* (e.g. Hessenauer et al. 2020. Nature Ecology & Evolution 4:626-638; Nigg et al. 2022. J. Fungi 8:637), since apples require little space, can be inoculated throughout the year (unlike elms) and are ideal for large-scale testing. When phenotyping mutants, GD apples facilitate initial large-scale screening and selection of individuals which will be inoculated to elm saplings for final validation.

L151: This is one of only two instances where cpm is used in the manuscript. Please include a description for this. It is probably not common knowledge for most readers.

We agree and we added the signification of cpm in the manuscript (L148-149).

L159: For DEG analyses, did you only compare 3dpi and 10dpi? I think it would have been interesting to see how gene expression in planta compared with *Ophiostoma* isolates in vitro. This would have given an opportunity to compare "baseline" or background gene expression levels for the genes of interest when the fungus is not interacting with a live plant host. Maybe these genes are only expressed when interacting with a live host? Do we know this to be true?

Due to budget constraints, we chose to assay *in planta* as many strains of fungi as feasible, excluding *in vitro* cultures for taxa and lineages. Therefore, we cannot directly determine baseline gene expression *in vitro*. Nonetheless, revised Table S5 now provides extra information on 90 *Ophiostoma* genes that were expressed solely *in planta*, either in previous work by Nigg et al. (2022) (Table S5, columns C and D) or the present study (Table S5, columns E to K).

L161: It looks like this is the first use of the acronym DEG in the manuscript in its current order. If this is the order that it will be when published, I would suggest describing DEG here for the reader.

We thank the reviewer for highlighting this mistake. We added the signification of DEG in the manuscript (L158).

L209: I would suggest rewording this sentence slightly to clarify that the gene expression of strain AME2 was more like that of OU than AME1. This is somewhat vague as it is currently

written. Rewording should make a clearer distinction between similar gene expression vs. close phylogenetic relationship.

The sentence has been modified to emphasize the differences in gene expression patterns (L204-206).

L233-234: Similar to my comment above, please clarify the first statement in this paragraph by saying something like "...orthologs in the CAZy database confirmed that gene expression in strain OQ strongly differed..." To make it clear that you are talking about differences in gene expression and not other characteristics.

The sentence was better reformulated (L229-232), we thank the reviewer for the suggestion.

L258: For those not familiar with WGCNA, it is not clear what the various colors/modules represent. I would suggest that you include a brief introductory sentence to this paragraph with some information so that the reader can better interpret your results.

We added a few sentences for better understanding (L256-262).

L326-328: When it is discovered that some genes expressed in trees inoculated with *Ophiostoma* spp. are also present in elms that were mock-inoculated with water, is it possible to remove these from the analysis since they are unlikely to be attributable to the pathogens? Maybe you already did this and its not in the paper? Or maybe I missed it? It would be interesting to discuss

Indeed, the expressed genes in the mock-inoculated control were excluded from PCA, Heatmap, GO, KEGG, CAZy, and PHI-base analyses, as indicated on "Overview of the *Ophiostoma* spp. transcriptomes in planta of the manuscript" (L140-142). However, these genes were included in the WGCNA analysis, and therefore explain why the grey module was significantly correlated with the water control treatment.

L345-350: Similar to my earlier comments regarding differences in gene expression among taxa-please include some terms in this paragraph to clarify that separation among these groups in the PCA plots reflects differences in gene expression. What does it mean for these groups to be closer together or further apart in the PCA plot? I know this seems obvious, but it will be easier for the reader to interpret if you include some context. Here is an example with some suggested edits: "Thus, strains ONU and OHU were consistently close to each other, indicating similar gene expression profiles. whereas strains AME1 and AME2 representing *O. novo-ulmi* ssp. *americana* lineages 1 and 2 were often further apart from each other than they were from other DED strains, suggesting that gene expression profiles within this lineage are highly variable.

The sentence has been reformulated (L346-350), we thank the reviewer for the suggestion.

L357-359: It might be good to provide some speculation as to why the deletion of this gene had this effect. It sounds like cerato-ulmin has an important function as hydrophobin, which serves

several biological roles according to the literature. Is there any biological reason why you think removing this gene would diminish pathogenicity? Any speculation as to why there is a dose-dependent effect when inoculating with this strain?

The *gpf1* gene encodes a transcription factor regulating growth, conidia germination, appressorium formation, and virulence in several fungal pathogens. As we mentioned in the Discussion, mutants for this gene obtained in other fungal pathogens exhibit multiple phenotypical changes, including loss of virulence. Therefore, we were not surprised that the ONU Δ *ogf1* mutant turned out to be less virulent than its wild-type H327 progenitor. We did not expect, however, that gene OnuG4296 encoding the hydrophobin cerato-ulmin (CU) would be silenced. Yet, we do not want to speculate on the individual contribution of CU gene silencing to the observed loss of virulence, since the global and CAZyme-gene transcriptomes of mutant Δ *ogf1* differed significantly from those from H327 (especially at 10 dpi). We believe that recovering targeted mutants for various *Ophiostoma* species and lineages would be required for properly assessing the biological role of CU.

As for the dose-dependent effect observed when inoculating strain ONU Δ *ogf1*, we now have experimental evidence that it is not specific to this strain. For instance, ongoing analyses of other insertional mutants (including mutants described in Nigg *et al.* 2022. *J. Fungi* 8:637) showed a similar effect for ONU mutant Δ *Aox1* 5-1 in which gene OnuG5955 encoding an alcohol oxidase was knocked out. We added this information in the last sentence of the paragraph (L359-360).

L364: This change in gene expression in genes related to carbohydrate metabolism seems significant. Can you provide some interpretation in a biological context? Any reason these would be upregulated at 10 dpi and not 3 dpi or 96 hpi? What biological functions of *Ophiostoma* spp. would rely on these gene products during this stage of the infection process? If there is any information on this in the existing literature, please cite.

We added one citation, and the text was slightly modified (L367-371).

L387-396: Was there any upregulation (or downregulation?) of genes related to CWDEs? Is there any evidence that *Ophiostoma* spp. produce these enzymes? Is cell wall degradation known to be important for *Ophiostoma* pathogenicity? I do not generally think of these organisms as causing any form of necrosis or enzymatic cell wall degradation, but maybe there is evidence for this elsewhere in the literature?

Yes, we observed differential expression of genes belonging to the GH6 and GH7 families, as indicated in the manuscript (L386-392 and Figure 5). Family GH6 is represented by gene OnuG0547 DEG at 3 dpi in OU and AME2, and DEG at 10 dpi in OHU. Family GH7 is represented by gene Onu5654 DEG at 3 dpi in AME2, and gene Onu0647 DEG in AME2 at 3 dpi and DEG in OU and OQ at 10 dpi. However, it is unclear whether DEG causes significant erosion of vessel walls despite the presence of DEGs in said families. The presence of these DEGs may have

triggered a robust immune response from the plant, resulting in the formation of tyloses and accumulation of other materials, which eventually blocked water transportation to both the plant and its leaves.

L413: Very interesting! Seems significant.

Indeed, which makes this gene a possible candidate for future functional analysis.

L428-430: It is not immediately clear how these results support the conclusion of "compensation". Can you provide more context here for the reader? If you have confirmed that the mutant $\Delta ogf1$ does not produce the gene product, then what are these other transcription factors compensating for?

This statement has been revised (L434-436). The term "compensation" should only be used after functional analysis has been conducted and the results properly analyzed. At that point, this hypothesis may be considered if it is relevant. Please note that we have indicated in the Materials and Methods section (L529-532) that ORF of *ogf1* is absent in the $\Delta ogf1$ mutant. Therefore, there are no RNA transcripts nor protein products from *ogf1* in the mutant.

L562: Which external symptoms were assessed specifically? Were there any measurements taken or was it purely qualitative? Please provide more information here.

The assessment of external foliar symptoms in the four inoculation trials is now presented with additional details in subsection "*Inoculation procedure and sampling of elms*" of the "Materials and Methods" section (L546-570).

Figure 7: If the numbers in the red, white, and blue boxes are meant to be interpreted by the reader, please increase the font size.

We provided a new figure with increased font size, and added "Pearson's rank correlation" to the figure.

Figure 8: Same as above. If text is meant to be interpreted by the reader, suggest increasing font size.

We provided a new figure with increased font size.

Reviewer #2 (Comments for the Author):

The manuscript untitled "Unraveling the transcriptional features and gene expression networks of pathogenic and saprotrophic *Ophiostoma* species during the infection of *Ulmus americana*" (control no. Spectrum03694-23) by de Oliveira and co-workers describes a transcriptomic comparative analysis of seven fungal plant pathogen strains, *Ophiostoma* sp., responsible for

Dutch elm disease. These strains include mutant strain and a saprotroph strain with low to no virulence. The results of this study are novel and interesting for the field due to the in planta setting of the experiment. In planta transcriptomic analysis is known to be more difficult due to complexity of the samples and resulting background noise. This study shows very good quality data validated by statistical tests.

The transcriptomic analysis presented here has been done at two time points: 3 days and 10 days after infection. As expected with this kind of analysis, a large number of genes show a differential expression among strains and conditions. This work is a good basis for further gene candidate functional characterization.

Page lines 125-130. Inoculum concentration on virulence, this paragraph is not clear to me. What was the purpose of testing several inoculum concentration? Is it to see the immune response? And, how many plants species were tested and why?

Different inoculum concentrations were tested to verify whether there might be a pathogen inoculum dose-dependent response of elm saplings. This was prompted by the observation that in previous studies (e.g. Bowden *et al.* 1996. *Mol. Plant. Microbe Interact.* 9:556–564; Temple *et al.* 2009. *N. Z. J. For. Sci.* 39:29–37; Nigg *et al.* 2022. *J. Fungi* 8:637), most ONU mutants for candidate pathogenicity genes remained highly virulent towards *U. americana*. We suspected that these studies relied on inoculum doses (typically over 50 000 spores per sapling) that might be too high given the high susceptibility of *U. americana* to DED. In the work reported in our submission, we verified pathogen inoculum dose-dependent response for wild-type strain ONU H327 and the OSCAR $\Delta ogf1$ mutant derived from it. Inoculations at low doses (12 500 spores or less per saplings) allowed us to conclude that the deletion of *ogf1* resulted in a significantly altered pathogenicity phenotype.

We slightly altered the relevant section in "Results" to specify that inoculum concentrations were only tested for strains H327 and $\Delta ogf1$. Readers will find technical details in the "Inoculation procedure and sampling of elms" of the "Materials and Methods" section. As indicated in our response to a question by Reviewer 1 about the dose-dependent effect, we also added relevant information in the Discussion (L354-360).

Page 9 line 184. OpKS8 putative gene cluster, what are the annotated function of these genes? Based on fig. S2, gene 7303 is also upregulated, is it part of the same OpKS8 gene cluster?

We added text (L186) referring interested readers to Table S6 for the annotation of the four genes from the Opks8 cluster that were upregulated at 10 dpi.

Gene OnuG7303 was not identified as a gene belonging to the OpKS8 cluster. However, results of WGCNA indicate that it has high connectivity with genes belonging to the cluster, as shown in figure 8.

Page 11, line 232, CAZyme-encoding genes. The Auxiliary Activity family AA (lignin/aromatic active enzymes) is not represented in figure 5. Are they missing from the analysis? What about the known regulators in other species such as Carbon repressor, cellulases, hemicellulases and pectin regulators?

Figure 5 shows all CAzyme-encoding genes that were DEG. Among them GH5, GH6, GH7, GH8 and GH45 are part of the cellulases. The AA family is represented by the GH61 subfamily. The GH15 family is represented by a xylan α -1,2-(4-O-methyl)-glucuronidase, one of the enzymes responsible for the degradation of xylan, a component of hemicellulose. The pectin lyase is represented by PL1.

A comment on the format, at first, I found difficult to understand and follow the results due to the use of abbreviations such as pdi, ONU, OU, AME1 etc. DED was defined in the introduction, but all the other abbreviations were only mentioned in the mat and met section (at the end of the manuscript). I understand it is a format issue but to ease the reading I suggest simply defining them in the introduction: example line 70-71, *Ophiostoma ulmi* (OU).

We included abbreviations for each organism in the introduction of the manuscript (L73-111)

There are a few spelling mistakes to check throughout the suppl. files both table titles and figures + text. (example table S4, hhighly).

We thank the reviewer for drawing our attention to this. We checked for typos and spelling mistakes throughout the revised manuscript and supplementary files, and hope we eliminated all of them.

Re: Spectrum03694-23R1 (Unraveling the transcriptional features and gene expression networks of pathogenic and saprotrophic *Ophiostoma* species during the infection of *Ulmus americana*)

Dear Miss Thais Campos de Oliveira:

Your manuscript has been accepted, and I am forwarding it to the ASM production staff for publication. Your paper will first be checked to make sure all elements meet the technical requirements. ASM staff will contact you if anything needs to be revised before copyediting and production can begin. Otherwise, you will be notified when your proofs are ready to be viewed.

Sincerely,
Lindsey Burbank
Editor
Microbiology Spectrum